# The formation of $K_V2.1$ macro-clusters is required for sex-specific differences in L-type $Ca_V1.2$ clustering and function in arterial myocytes

Collin Matsumoto [1], Samantha C. O'Dwyer[1], Declan Manning [1], Gonzalo Hernandez-Hernandez[1], Paula Rhana [1], Zhihui Fong [1], Daisuke Sato[2], Colleen E. Clancy[1], Nicholas C. Vierra [1], James S. Trimmer [1] & L. Fernando Santana [1✉]

In arterial myocytes, the canonical function of voltage-gated $Ca_V1.2$ and $K_V2.1$ channels is to induce myocyte contraction and relaxation through their responses to membrane depolarization, respectively. Paradoxically, $K_V2.1$ also plays a sex-specific role by promoting the clustering and activity of $Ca_V1.2$ channels. However, the impact of $K_V2.1$ protein organization on $Ca_V1.2$ function remains poorly understood. We discovered that $K_V2.1$ forms micro-clusters, which can transform into large macro-clusters when a critical clustering site (S590) in the channel is phosphorylated in arterial myocytes. Notably, female myocytes exhibit greater phosphorylation of S590, and macro-cluster formation compared to males. Contrary to current models, the activity of $K_V2.1$ channels seems unrelated to density or macro-clustering in arterial myocytes. Disrupting the $K_V2.1$ clustering site ($K_V2.1_{S590A}$) eliminated $K_V2.1$ macro-clustering and sex-specific differences in $Ca_V1.2$ cluster size and activity. We propose that the degree of $K_V2.1$ clustering tunes $Ca_V1.2$ channel function in a sex-specific manner in arterial myocytes.

[1] Department of Physiology and Membrane Biology, School of Medicine, University of California, Davis, CA, USA. [2] Department of Pharmacology, School of Medicine, University of California, Davis, CA, USA. ✉email: lfsantana@ucdavis.edu

Activation of dihydropyridine-sensitive, voltage-gated L-type $Ca_V1.2$ channels plays a crucial role in the development of myogenic tone[1], a process of autoregulation that enables arteries to regulate their diameter in response to changes in intravascular pressure[2]. This mechanism, independent of neural or hormonal influences, is critical to maintain constant blood flow despite changes in blood pressure.

The current model of the myogenic response proposes that mechanical stretch of the membrane leads to the activation of TRPC6[3], TRPM4[4], and TRPP1 (PKD2)[5] channels, which depolarizes arterial myocytes, activating $Ca_V1.2$ channels[1]. Activation of a single or a small cluster of $Ca_V1.2$ channels results in a local increase in intracellular $Ca^{2+}$ concentration ($[Ca^{2+}]_i$) termed a "$Ca_V1.2$ sparklet"[6–8]. Summation of multiple $Ca_V1.2$ sparklets leads to a global increase in $[Ca^{2+}]_i$ that triggers muscle contraction.

$Ca_V1.2$ channels form clusters in the plasma membrane via a stochastic self-assembly mechanism[9]. $Ca_V1.2$ channels within these clusters can gate cooperatively in response to $Ca^{2+}$-driven physical interactions of adjacent channels[10,11]. $Ca_V1.2$ channels in this configuration allow for larger $Ca^{2+}$ influx compared to random, independent openings of individual channels. In vascular smooth muscle, cooperative gating of $Ca_V1.2$ channels has been estimated to contribute up to ~50% of $Ca^{2+}$ influx during the development of myogenic tone[12].

One route of negative feedback regulation of membrane depolarization and $Ca^{2+}$ influx via $Ca_V1.2$ channels occurs through the depolarization-mediated activation of delayed rectifier voltage-gated $K_V2.1$ channels[13,14]. In its canonical role, $K_V2.1$ proteins in arterial smooth muscle cells form ion conducting voltage-gated $K^+$ channels whose activation results in membrane potential hyperpolarization, thereby affecting myocyte $[Ca^{2+}]_i$ and myogenic tone[13]. Until recently, the accepted role of $K_V2.1$ protein in arterial myocytes was to form $K^+$ conducting channels. However, our recent work suggests that only about 1% of the $K_V2.1$ channels in the plasma membrane of arterial smooth muscle are conductive[15]. Indeed, a growing body of evidence suggests that $K_V2.1$ proteins have dual conductive and structural roles in the surface membrane of smooth muscle cells and neurons[15–17].

In neurons and HEK293T cells, $K_V2.1$ is expressed in large macro-clusters[16,18–23]. A 26 amino acid region within the C-terminus of the channel called the proximal restriction and clustering (PRC) domain was determined to be responsible for this expression pattern[24]. The high-density clustering of $K_V2.1$ channels is influenced by phosphorylation of serine residues within the PRC domain[25–27]. Additionally, in heterologous expression systems, the majority of $K_V2.1$ channels within macro-clusters are considered non-conductive[18,19,28]. Little to no channel activity was detectable within $K_V2.1$ clusters, whereas currents could be recorded in areas with diffuse $K_V2.1$ expression[19]. One of the structural roles of $K_V2.1$ is to promote clustering of $Ca_V1.2$ channels, thus increasing the probability of $Ca_V1.2$-to-$Ca_V1.2$ interactions within these clusters[16,17].

Both, the conductive and structural roles of $K_V2.1$, depend on the level of expression of this protein, which in arterial myocytes varies with sex[15]. In female myocytes, where expression of $K_V2.1$ protein is higher than in male myocytes, $K_V2.1$ has both conductive and structural roles. Female myocytes have larger $Ca_V1.2$ clusters, $[Ca^{2+}]_i$, and myogenic tone than male myocytes. In contrast, in male myocytes, $K_V2.1$ channels regulate membrane potential, but not $Ca_V1.2$ channel clustering.

Based on these data, a model was proposed in which $K_V2.1$ function varies with sex[15]. In males, $K_V2.1$ channels primarily control membrane potential, but in female myocytes $K_V2.1$ plays dual electrical and $Ca_V1.2$ clustering roles. Currently, it is unclear whether the conductive and structural functions of $K_V2.1$ in native arterial myocytes rely on its clustering ability, and if this relationship is sex-dependent.

In this study, we tested the hypothesis that conductive and structural roles of $K_V2.1$ channels in male and female arterial myocytes depend on their capacity to form clusters in studies of wild-type (WT) and S586A mutant rat $K_V2.1$ channels expressed in heterologous cells, and in arterial myocytes from a novel gene edited knock-in mouse expressing the S590A mutation. We focused on serine 586 within the PRC domain (amino acids 573-598) of rat $K_V2.1$ because a point mutation changing this amino acid to a non-phosphorylatable alanine decreases the $K_V2.1$ clustering phenotype[24]. This corresponds to a S590A point mutation in the mouse $K_V2.1$ channel. Our data show that $K_V2.1$ is expressed into large macro-clusters composed of micro-clusters that can only be resolved with super-resolution microscopy. The $K_V2.1_{S586A}$ point mutation nearly eliminated $K_V2.1$ macro-clusters but had a minimal impact on micro-clusters. Notably, we find that $K_V2.1$ channel function is not dependent on its ability to form macro-clusters in arterial myocytes of either sex. Rather, $K_V2.1$ macro-clustering enhances $Ca_V1.2$ channel clusters and activity. Based on these results, we propose a new model in which macro-clustering of $K_V2.1$ in arterial myocytes alters $Ca_V1.2$ channel organization and function in a sex-specific manner but has no impact on its conductive function.

## Results

**$K_V2.1$ macro-clusters are composed of micro-clusters of $K_V2.1$ and are declustered by the $K_V2.1_{S586A}$ mutation.** We began our study by determining the spatial distribution in heterologous cells of exogenously expressed wild-type rat $K_V2.1$ ($K_V2.1_{WT}$) channels and $K_V2.1$ channels in which serine at position 586 was mutated to an alanine ($K_V2.1_{S586A}$) using confocal and super-resolution ground state depletion (GSD) microscopy (Fig. 1). Both channels were tagged at their N-terminus with the red-shifted fluorescent protein mScarlet.

Figure 1a shows confocal maximum intensity projection images of 3D reconstructions of representative HEK293T cells expressing $K_V2.1_{WT}$ and $K_V2.1_{S586A}$. To the right of each image, we show single plane images from the bottom and center of each cell. Figure 1b–e shows a quantitative analysis of the number and volume of $K_V2.1$ clusters from these 3D confocal images. The frequency distributions of $K_V2.1_{WT}$ (black) and $K_V2.1_{S586A}$ (blue) cluster volumes could be fit with a single exponential function. Of note, the number of clusters were larger in most volume bins for the $K_V2.1_{S586A}$ mutation compared to $K_V2.1_{WT}$. For example, in the same number of cells ($n = 11$), we detected a total of 25,174 $K_V2.1_{WT}$ clusters and 31,893 $K_V2.1_{S586A}$ clusters. The number of clusters per cell was $2286 \pm 188.4$ in $K_V2.1_{WT}$ (median = 2227 clusters) and $2899 \pm 259.9$ clusters in $K_V2.1_{S586A}$ (median = 2889 clusters) ($P = 0.04$) (Fig. 1c). The total cluster volume per cell of $K_V2.1_{WT}$ was $254.5 \pm 20.8$ $\mu m^3$ (median = 246.0 $\mu m^3$) compared to $192.5 \pm 23.5$ $\mu m^3$ in cells expressing $K_V2.1_{S586A}$ (median = 149.0 $\mu m^3$) ($P = 0.03$) (Fig. 1d). Interestingly, the mean cluster volumes were larger between $K_V2.1_{WT}$ channels at $0.12 \pm 0.01$ $\mu m^3$ (median = 0.12 $\mu m^3$) and $K_V2.1_{S586A}$ at $0.06 \pm 0.01$ $\mu m^3$ (median = 0.06 $\mu m^3$) ($P = <0.0001$) (Fig. 1e). These data suggest that $K_V2.1_{WT}$ channels are expressed into clusters and that $K_V2.1_{S586A}$ expression is more diffuse and exhibits a more uniform distribution.

Analysis of our confocal microscopy data showed that in HEK293T cells, $K_V2.1_{WT}$ is expressed in clusters of different sizes but do in fact form large macro-clusters. Using this confocal microscopy analysis, we sought to define the size of a $K_V2.1$ macro-cluster. We began with the mean cluster volume of

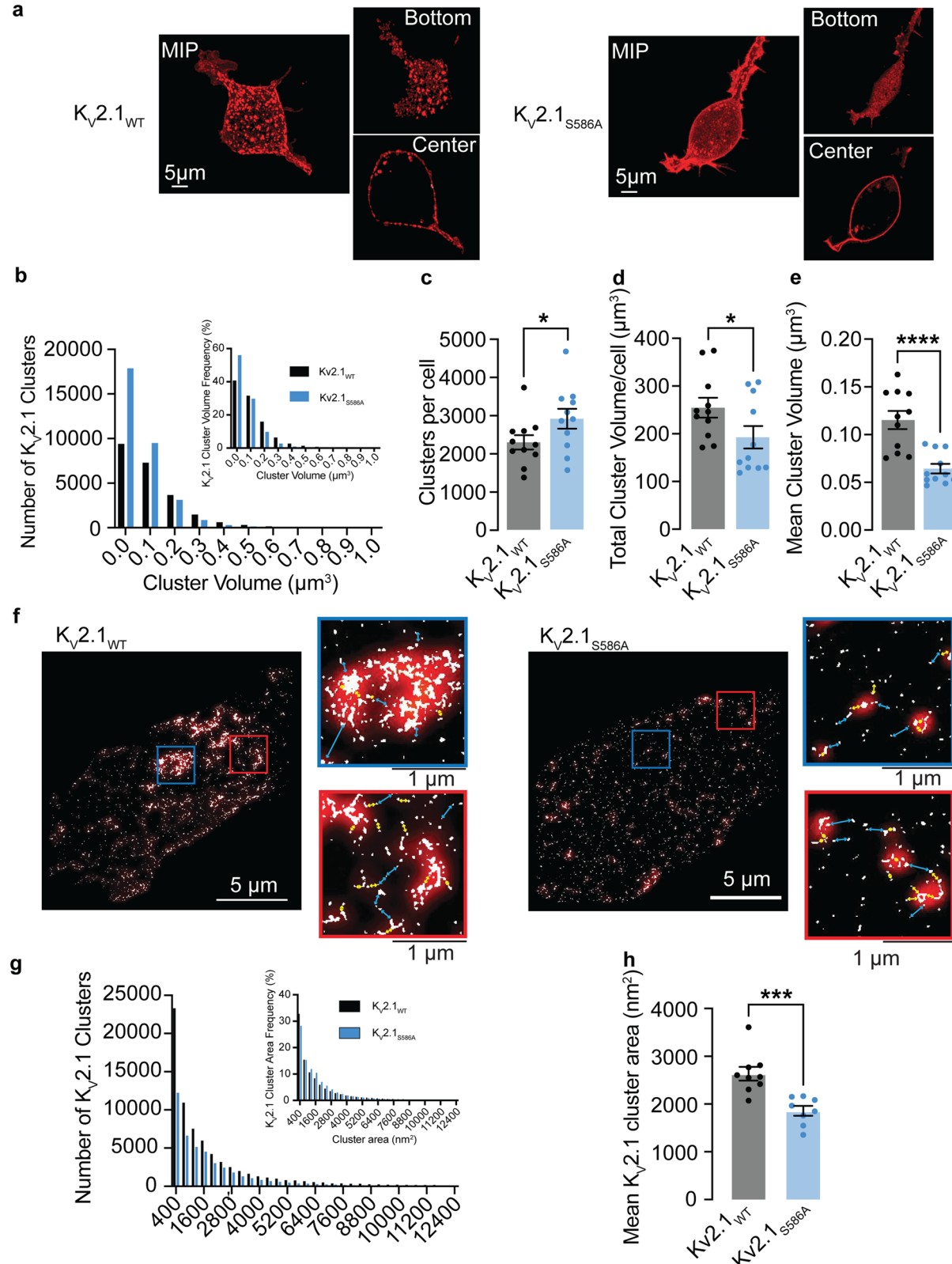

$K_V2.1_{WT}$ generated in Fig. 1e. Our analysis provided a mean cluster volume of 0.12 μm³. Assuming the volume of a cluster is spherical, we extrapolated the diameter of the macro-cluster to be 278 nanometers (nm). The standard deviation of these measurements was 0.03. Thus, the mean minus 2 standard deviations provides a lower end limit with a 95% confidence and aligned with the lateral resolution of our confocal microscopy.

Accordingly, we define the lower limit of a macro-cluster as a cluster that is larger than 0.06 μm³ or 193 nm in diameter, with clusters smaller than this classified as micro-clusters.

Figure 1f provides representative super-resolution ground-state depletion (GSD) TIRF images (lateral resolution ≈ 40 nm) from representative cells expressing $K_V2.1_{WT}$ or $K_V2.1_{S586A}$. Note that data are provided on an area basis since we are imaging a single

**Fig. 1 K$_V$2.1 macro-clusters are declustered into micro-clusters with the K$_V$2.1$_{S586A}$ point mutation. a** Confocal maximum intensity projection images of HEK293T cells transfected with K$_V$2.1$_{WT}$ or K$_V$2.1$_{S586A}$ tagged with mScarlet. Insets show single plane images of the bottom or center of each cell. **b** Number of K$_V$2.1 cluster volumes of K$_V$2.1$_{WT}$ (black) and K$_V$2.1$_{S586A}$ (blue) in transfected HEK293T cells. Insets quantify relative frequency histograms as a percentage of K$_V$2.1 volumes. Plots representing (**c**) number of clusters per cell, (**d**) total cluster volume per cell, and (**e**) mean cluster volumes ($n = 11$ K$_V$2.1$_{WT}$ and 11 K$_V$2.1$_{S586A}$ cells). **f** Representative super-resolution GSD images of immunolabeled K$_V$2.1 channels in transfected HEK293T cells. Insets show 4 μm$^2$ regions of interest. Red overlay depicts Gaussian blur. Cyan lines indicate distances of greater than 200 nm while yellow arrows represent distances less than 200 nm. **g** Number of clusters of K$_V$2.1$_{WT}$ (black) and K$_V$2.1$_{S586A}$ (blue) by cluster area in transfected HEK293T cells. Insets quantify relative frequency histograms as a percentage of K$_V$2.1 areas. **h** Summary plot of mean K$_V$2.1 cluster areas from super-resolution images ($n = 9$ K$_V$2.1$_{WT}$ and 8 K$_V$2.1$_{S586A}$ cells). $*P < 0.05$, $**P < 0.01$, $***P < 0.001$, $****P < 0.0001$. Error bars indicate mean ± SEM.

plane footprint of the cell in contrast to capturing multiple Z-slices. We also provide two regions of interest by each image. To increase our confidence in this observation, we utilized a Gaussian blur (shown in red) to decrease the resolution of the GSD signal to match that of confocal microscopy. Cyan arrows indicate distances greater than 200 nm between signals produced by GSD, while yellow arrows indicate distances less than 200 nm. Notably, our GSD image with a Gaussian blur accurately reproduced the clustering phenotype of K$_V$2.1$_{WT}$ at the confocal level, providing further evidence that macro-clusters are composed of micro-clusters. Although we do not observe the large macro-clusters in K$_V$2.1$_{S586A}$ expressing cells, we can still resolve K$_V$2.1 micro-clusters.

K$_V$2.1$_{WT}$ and K$_V$2.1$_{S586A}$ channels are organized into clusters of varied sizes (Fig. 1g), and consistent with the lower resolution confocal data, the distribution of K$_V$2.1$_{S586A}$ channel clusters was more diffuse than that of K$_V$2.1$_{WT}$. It should be noted that the regions of interest (ROI) of our GSD images reveal that the macro-clusters observed at the confocal level are, in fact, made up of numerous micro-clusters. The frequency distribution of the cluster areas of K$_V$2.1$_{WT}$ and K$_V$2.1$_{S586A}$ obtained from GSD imaging could both be fit with an exponential function (Fig. 1g). The mean cluster area of K$_V$2.1$_{WT}$ was $2634 \pm 143$ nm$^2$ (median $= 2568$ nm$^2$), larger than that of K$_V$2.1$_{S586A}$ channels which was $1860 \pm 104$ nm$^2$ (median $= 1907$ nm$^2$) ($P = 0.0003$) (Fig. 1h). This is likely due to the absence of larger clusters in cells expressing K$_V$2.1$_{S586A}$. Notably, cluster density was $41 \pm 5$ clusters per micron and $29 \pm 4$ clusters per micron for K$_V$2.1 and K$_V$2.1$_{S586A}$, respectively.

Our finding that the size distributions of K$_V$2.1$_{WT}$ and K$_V$2.1$_{S586A}$ clusters could be described by exponential functions, the hallmark of a Poisson process, suggests that these clusters are formed stochastically[29]. To test this hypothesis, we implemented the stochastic modeling approach employed by Sato et al.[9] to determine whether we could reproduce our cluster distributions and make testable predictions regarding plasma membrane protein organization. As shown in Supplementary Fig. 1a, b, our stochastic self-assembly model effectively reproduced the steady-state size distributions that we measured for K$_V$2.1$_{WT}$ and K$_V$2.1$_{S586A}$ proteins embedded in the surface membrane of HEK293T cells. The parameters used in the model are summarized in Supplementary Fig. 1c. These in silico data suggest that in HEK293T cells, K$_V$2.1$_{WT}$ has a higher probability of nucleation and cluster growth than K$_V$2.1$_{S586A}$ channels. The probability of channel removal was similar.

**K$_V$2.1$_{S590A}$ mutation does not affect overall K$_V$2.1 channel expression but declusters smooth muscle K$_V$2.1 macro-clusters in a sex-specific manner.** Next, we investigated whether, as in heterologous expression system (i.e., Fig. 1), K$_V$2.1 channels cluster in arterial myocytes and whether this clustering could be disrupted via mutation of critical amino acids in the PRC domain. To do this, we used CRISPR/Cas gene editing to generate a knock-in mouse line expressing the S590A point mutation,

corresponding to the S586A mutation in rat K$_V$2.1, at the KCNB1 locus of a C57BL/6 J mouse (see Methods section for a full description of how these mice were generated).

We isolated, fixed, permeabilized, and labeled arterial myocytes from both male and female K$_V$2.1$_{WT}$ and K$_V$2.1$_{S590A}$ knock-in mice. We double labeled with wheat germ agglutinin-488 (WGA488) to identify the sarcolemma and K$_V$2.1-specific antibodies in male and female K$_V$2.1$_{WT}$ and K$_V$2.1$_{S590A}$ myocytes using confocal microscopy.

Figure 2a shows representative 3D images of fixed male mesenteric smooth muscle cells labeled with WGA488 and immunolabeled for K$_V$2.1. We investigated whether the K$_V$2.1$_{S590A}$ mutation leads to altered expression levels of the channel in arterial myocytes. Our analysis showed that total, cell-wide K$_V$2.1-associated fluorescence normalized to cell membrane area was similar in sex-matched K$_V$2.1$_{WT}$ and K$_V$2.1$_{S590A}$ myocytes (Fig. 2b), suggesting that the K$_V$2.1 expression is similar in these cells.

Further analysis of K$_V$2.1 clusters was restricted to those that overlapped with the WGA-mapped plasma membrane. In both, the K$_V$2.1$_{WT}$ and K$_V$2.1$_{S590A}$ males, the frequency distribution of K$_V$2.1 cluster sizes were similar in terms of relative values in all volume bins and could be fit with an exponential decay function (Fig. 2c). The mean cluster volume in K$_V$2.1$_{WT}$ males was $0.07 \pm 0.01$ μm$^3$ (median $= 0.07$ μm$^3$) compared to a mean of $0.07 \pm 0.01$ μm$^3$ (median $= 0.06$ μm$^3$) in K$_V$2.1$_{S590A}$ male myocytes. ($P = 0.34$) (Fig. 2d). Additionally, total clusters per cell of $551.6 \pm 71.1$ clusters, (median $= 505$ clusters) in K$_V$2.1$_{WT}$ male myocytes were not significantly different from total clusters per cell of $444.2 \pm 33.4$ clusters (median $= 419$ clusters) in K$_V$2.1$_{S590A}$ males ($P = 0.10$) (Fig. 2e). The percentage of the membrane occupied by clusters in K$_V$2.1$_{WT}$ male myocytes was on average $5.5 \pm 0.9\%$ (median $= 5.1\%$), similar to the average in K$_V$2.1$_{S590A}$ males of $5.0 \pm 1.0\%$ (median $= 4.6\%$) ($P = 0.34$) (Fig. 2f).

We next investigated whether the K$_V$2.1$_{S590A}$ mutation leads to altered expression levels of the channel in female arterial myocytes. Figure 2g shows representative 3D images of fixed female mesenteric smooth muscle cells labeled with WGA488 and immunolabeled for K$_V$2.1. Similar to male myocytes, our analysis revealed that the total K$_V$2.1-associated fluorescence, normalized to the cell membrane area, exhibited no significant difference between sex-matched K$_V$2.1$_{WT}$ and K$_V$2.1$_{S590A}$ myocytes (Fig. 2h).

In sharp contrast to male myocytes, K$_V$2.1$_{S590A}$ female myocytes exhibited an increased proportion of smaller clusters as compared to those from K$_V$2.1$_{WT}$ females (Fig. 2i). Accordingly, mean cluster size of K$_V$2.1$_{WT}$ in female myocytes was $0.14 \pm 0.10$ μm$^3$ (median $= 0.14$ μm$^3$), significantly larger than mean cluster size of $0.10 \pm 0.01$ μm$^3$ (median $= 0.09$ μm$^3$) ($P = 0.007$) in K$_V$2.1$_{S590A}$ females (Fig. 2j). K$_V$2.1$_{WT}$ female myocytes had $796 \pm 67$ clusters per cell, (median $= 815$ clusters) compared to only $422 \pm 34$ clusters per cell (median $= 415$ clusters) in K$_V$2.1$_{S590A}$ female myocytes ($P < 0.0001$) (Fig. 2k). Similarly, the percentage of the plasma membrane occupied by

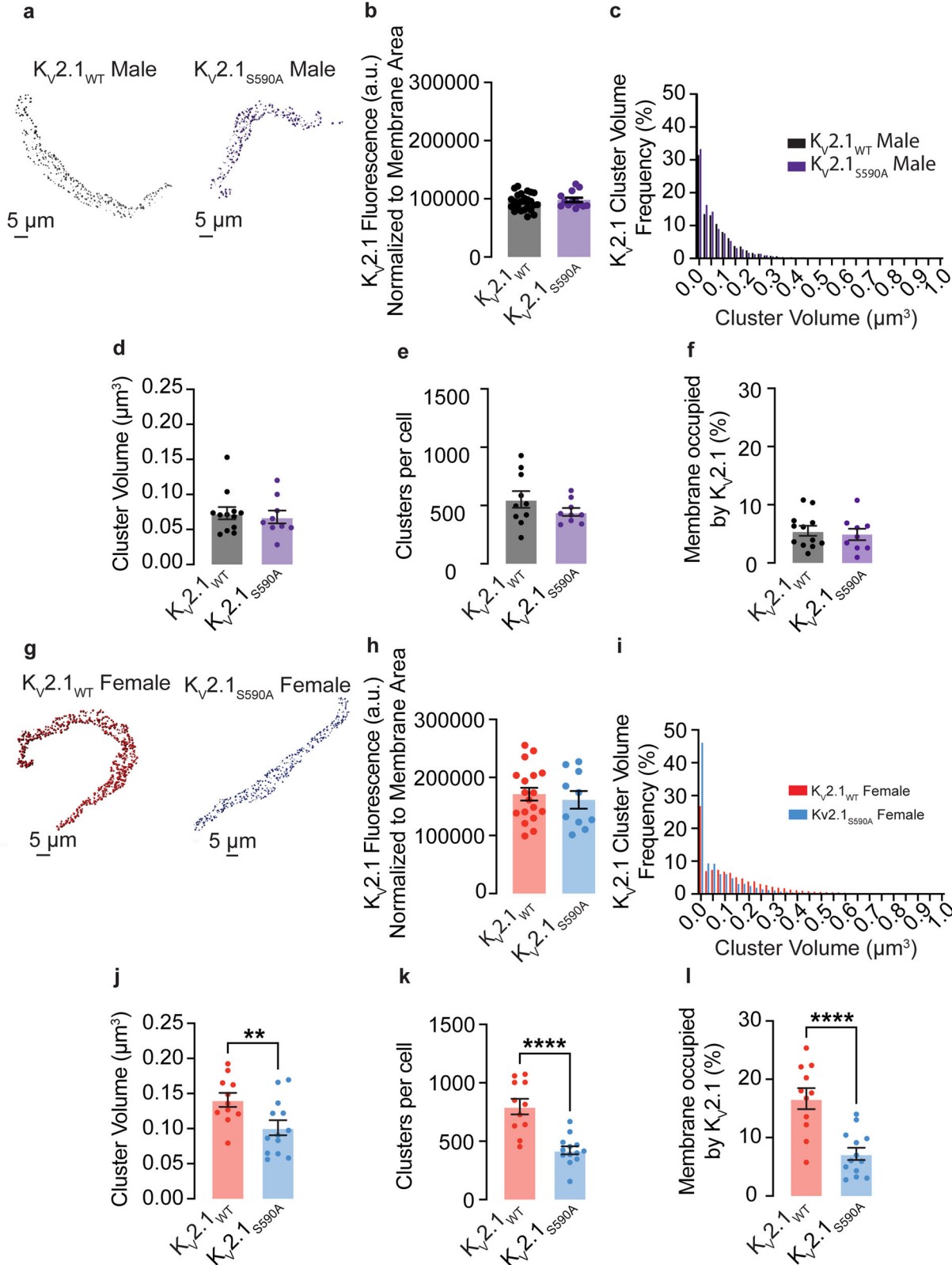

$K_V2.1$ was higher in $K_V2.1_{WT}$ female myocytes at $16.7 \pm 1.8\%$ (median $= 17.7\%$) in contrast to $K_V2.1_{S590A}$ female myocytes in which $K_V2.1$ clusters occupied on average $7.2 \pm 1.0\%$ of the plasma membrane (median $= 6.0$) ($P < 0.0001$) (Fig. 2l). The significantly lower $K_V2.1$ clustering profile in all metrics measured indicates that unlike in males, the S590A mutation decreases channel clustering in female myocytes.

Using the threshold set from our confocal imaging (i.e., macro-clusters are $>0.06 \, \mu m^3$), we quantified the number of macro-clusters expressed in myocytes from $K_V2.1_{WT}$ and $K_V2.1_{S590A}$ mice. Around 41% of clusters in $K_V2.1_{WT}$ male myocytes were identified as macro-clusters, with a similar percentage of 35% observed in samples from males with the $K_V2.1_{S590A}$ mutation. Approximately 57% of $K_V2.1$ clusters in $K_V2.1_{WT}$ female

**Fig. 2 The $K_V2.1_{S590A}$ mutation declusters $K_V2.1$ macro-clusters in arterial smooth muscle in a sex-specific manner. a** Representative maximum projection images of $K_V2.1$ channel clusters at the surface membrane of $K_V2.1_{WT}$ and $K_V2.1_{S590A}$ male myocytes. **b** Quantification of immunofluorescence labeled $K_V2.1$ normalized to cell membrane area in male myocytes ($n = 24$ $K_V2.1_{WT}$ and 15 $K_V2.1_{S590A}$ male myocytes). **c** Relative frequency as a percentage of $K_V2.1$ cluster volumes from $K_V2.1_{WT}$ male and $K_V2.1_{S590A}$ male myocytes. Summary data of $K_V2.1$ clusters in myocytes showing (**d**) mean cluster volumes, (**e**) clusters per cell, and (**f**) percent of the surface membrane occupied by $K_V2.1$ channels in male myocytes ($n = 12$ $K_V2.1_{WT}$ male cells and 9 $K_V2.1_{S590A}$ male cells). **g** Representative maximum projection images of $K_V2.1$ channel clusters at the surface membrane of $K_V2.1_{WT}$ and $K_V2.1_{S590A}$ female myocytes. **h** Quantification of immunofluorescence labeled $K_V2.1$ normalized to cell membrane area in female myocytes ($n = 18$ $K_V2.1_{WT}$ and 11 $K_V2.1_{S590A}$ female myocytes). **i** Relative frequency as a percentage of $K_V2.1$ cluster volumes from $K_V2.1_{WT}$ female and $K_V2.1_{S590A}$ female myocytes. Summary data of $K_V2.1$ clusters in myocytes showing (**j**) mean cluster volumes, (**k**) clusters per cell, and (**l**) percent of the surface membrane occupied by $K_V2.1$ channels in female myocytes ($n = 11$ $K_V2.1_{WT}$ and 13 $K_V2.1_{S590A}$ female cells). *$P < 0.05$, **$P < 0.01$, ***$P < 0.001$, ****$P < 0.0001$. Error bars indicate mean ± SEM.

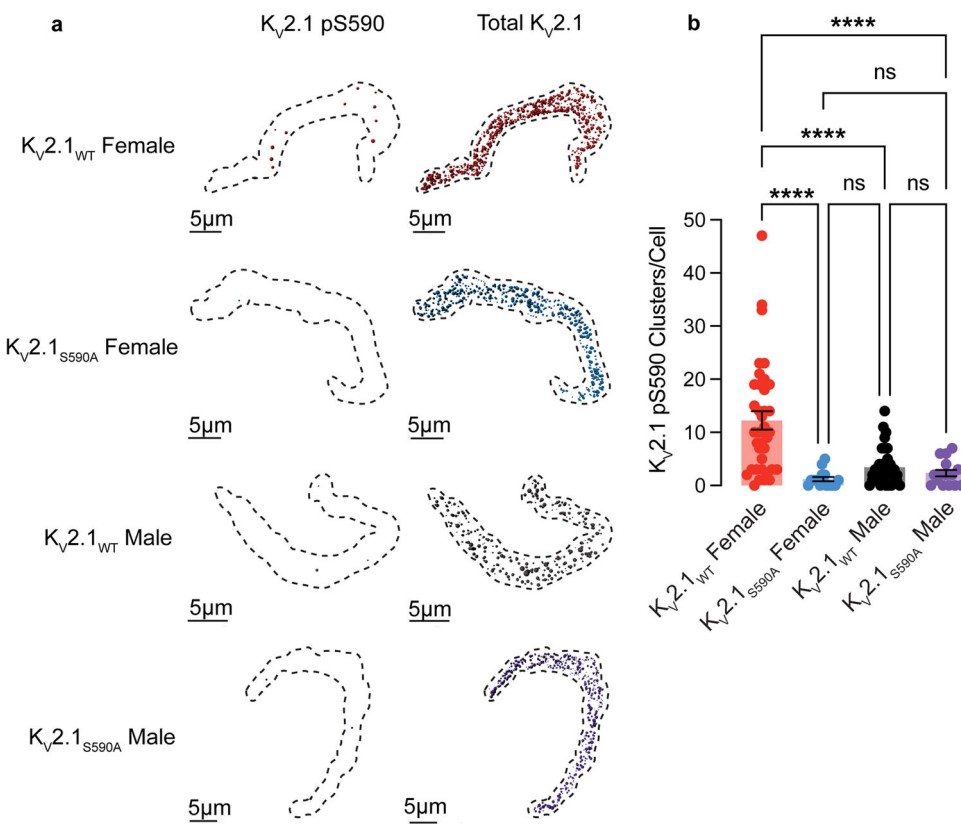

**Fig. 3 $K_V2.1_{WT}$ female myocytes exhibit more extensive $K_V2.1$ pS590 phosphorylation than those from $K_V2.1_{WT}$ males. a** Representative maximum projection images of immunolabeling for pS590 $K_V2.1$ and total $K_V2.1$channel clusters at the surface membrane of $K_V2.1_{WT}$ female (red), $K_V2.1_{S590A}$ female (blue), $K_V2.1_{WT}$ male (black), and $K_V2.1_{S590A}$ male (purple) cells. **b** Summary data of pS590 $K_V2.1$ clusters per cell ($n = 37$ $K_V2.1_{WT}$ male, 15 $K_V2.1_{S590A}$ male, 34 $K_V2.1_{WT}$ female and 15 $K_V2.1_{S590A}$ myocytes). *$P < 0.05$, **$P < 0.01$, ***$P < 0.001$, ****$P < 0.0001$. Error bars indicate mean ± SEM.

myocytes were classified as macro-clusters. Remarkably, in $K_V2.1_{S590A}$ female myocytes, macro-clusters accounted for approximately 50% of the total $K_V2.1$ clusters.

We also quantified $K_V2.1$ micro-clusters. Although the proportion of micro-clusters was similar in male $K_V2.1_{WT}$ and $K_V2.1_{S590A}$ myocytes (59% and 65%, respectively), female $K_V2.1_{S590A}$ myocytes exhibited a much larger proportion of micro-clusters (50%) compared to myocytes from $K_V2.1_{WT}$ females (42%). Hence, it can be reasoned that the S590A mutation has a sex-specific effect of reducing the extent of $K_V2.1$ macro-clustering in female but not male arterial myocytes without impacting channel expression.

**$K_V2.1$ S590 phosphorylation is higher in myocytes from female versus male $K_V2.1_{WT}$ mice.** To investigate the potential role of the S590 phosphorylation site in the sex-specific differences in $K_V2.1$ clustering, we conducted immunocytochemistry analyses

on arterial myocytes (Fig. 3). We tested the hypothesis that $K_V2.1_{WT}$ female myocytes exhibit a higher degree of $K_V2.1$ S590 phosphorylation compared to males, which could contribute to the observed sex-specific variations in $K_V2.1$ clustering. Accordingly, we utilized a monoclonal antibody (mAb L100/1[30]) specific for $K_V2.1$ that is phosphorylated at serine 590 (pS590).

Figure 3a shows cells double immunolabeled for pS590 and total $K_V2.1$ using confocal microscopy. One advantage of our study is that our $K_V2.1_{S590A}$ mice serve as an ideal negative control. As expected, $K_V2.1_{S590A}$ males exhibited 2.3 ± 0.6 clusters (median = 2.0 clusters) per cell and $K_V2.1_{S590A}$ females exhibited 1.2 ± 0.4 clusters (median = 1.0 clusters) per cell confirming the specificity of this mAb for $K_V2.1$ phosphorylated at S590. Remarkably, phosphorylated $K_V2.1$ clusters were observed in $K_V2.1_{WT}$ females (Fig. 3b) exhibiting on average 12.2 ± 1.7 of phosphorylated clusters (median = 10.0 clusters) per cell, whereas $K_V2.1_{WT}$ males exhibited 3.4 ± 0.6 clusters (median = 2.5 clusters) per cell. Collectively, these findings suggest that basal levels

of $K_V2.1$ pS590 phosphorylation are higher in $K_V2.1_{WT}$ females, which could account for their increased clustering of $K_V2.1$. Furthermore, these data support the notion that $K_V2.1$ phosphorylation in $K_V2.1_{WT}$ male myocytes is constitutively low, making the S590A mutation functionally indistinguishable from non-phosphorylated $Kv2.1_{WT}$ and thus ineffective in altering $K_V2.1$ clustering in male myocytes.

**Expression of clustering impaired $K_V2.1_{S590A}$ does not affect channel activity in arterial myocytes.** Three studies, one using Xenopus oocytes[28], one using HEK293T cells[19] and another from our group using arterial myocytes[15] have suggested that the vast majority of $K_V2.1$ channels (i.e., 98–99%) expressed in the plasma membrane of these cells are non-conductive. O'Connell et al.[19] suggested that, at least in HEK293T cells, $K_V2.1$ channel activity depends on their density and that channels within large, dense macro-clusters are non-conductive. A testable hypothesis raised by these data is that a larger fraction of $K_V2.1_{S590A}$ should be conductive and hence the amplitude of $K_V2.1$ currents in native arterial myocytes should differ between cells from $K_V2.1_{WT}$ and $K_V2.1_{S590A}$ mice (Fig. 4).

We tested this hypothesis using a multipronged approach. First, we used a mathematical modeling approach[31] to determine the predicted changes in macroscopic $K_V2.1$ currents with varied levels of functional channels (i.e., 0.1, 1, 10, 50, or 100%) in male and female arterial myocytes (Fig. 4a–d). The rationale for this analysis is that it provides a set of potential outcomes that can provide insights into the degree of $K_V2.1$ declustering in $K_V2.1_{S590A}$ myocytes. This model incorporated data (e.g., voltage-dependencies and number of channels in the sarcolemma) from O'Dwyer et al.[15].

As shown in Fig. 4a, the model predicts that with 100%, 50%, or 10% functional $K_V2.1$ channels in male myocytes would produce current densities at +50 mV of 7,006, 3,503, and 701 pA/pF, respectively. By contrast, at the same voltage, the in silico female arterial myocytes produce current densities of 17,293, 8,646, and 1,729 pA/pF with 100%, 50%, or 10% functional $K_V2.1$ channels (Fig. 4b). We also simulated the current-voltage relationships in male (Fig. 4c) and female (Fig. 4d) myocytes assuming 1% and 0.1% of $K_V2.1$ channels are conductive, which are more within the range with previous experimental results in heterologous systems[19,28] and native cells[15]. The magnitude of in silico $K_V2.1$ current densities with 1% or 0.1% functional channels was 70.1 and 5.57 pA/pF in male myocytes and 173 and 16.7 pA/pF in female myocytes.

Next, we recorded voltage-gated $K^+$ ($K_V$) currents in male and female $K_V2.1_{WT}$ and $K_V2.1_{S590A}$ arterial myocytes in response to 500 ms depolarizations to voltages between −50 and +50 mV before and after applying the $K_V2.1$ blocker RY785 (1 μM)[32,33]. This compound decreases $K_V2.1$ currents by blocking the pore of these channels[32] rather than by immobilizing their voltage sensor, as stromatoxin does[34]. As a first step in these experiments, we tested the specificity of the RY785 by recording $K_V$ currents before and after the application of this molecule in male and female $K_V2.1_{WT}$ and $K_V2.1$ null ($K_V2.1^{-/-}$) myocytes (Supplementary Fig. 2a–d). Notably, application of 1 μM RY785 decreased the amplitude of $K^+$ currents in $K_V2.1_{WT}$ but not in $K_V2.1^{-/-}$ myocytes of either sex. This indicates that RY785 is a specific blocker of $K_V2.1$ channels in arterial myocytes.

Having completed these critical control experiments, we recorded $K_V$ currents from $K_V2.1_{WT}$ and $K_V2.1_{S590A}$ myocytes. We noted that the amplitude of the composite K currents were similar in myocytes from $K_V2.1_{S590A}$ mice compared to myocytes from sex-matched $K_V2.1_{WT}$ littermates (Supplementary Figs. 2e, f).

Importantly, for both sexes, RY785-sensitive $K_V2.1$ currents were also similar in male (Fig. 4e) and female (Fig. 4f) $K_V2.1_{WT}$ and $K_V2.1_{S590A}$ myocytes. Indeed, a comparison of the experimental and in silico amplitudes of the macroscopic $K_V2.1$ currents suggests that less than 1% of the channels are functional in myocytes from both male and female $K_V2.1_{WT}$ and $K_V2.1_{S590A}$ mice. When taken together with our analyses of $Kv2.1$ clustering detailed above, these findings suggest that in arterial myocytes $K_V2.1$ channel activity is not determined by the extent and nature of its clustering.

**The $K_V2.1_{S590A}$ mutation diminishes $K_V2.1$ and $Ca_V1.2$ proximity in female myocytes.** We used the proximity ligation assay (PLA) to interrogate the impact of the $K_V2.1_{S590A}$ mutation on protein-protein proximity, at a resolution of approximately 50 nm[35,36] (Supplementary Fig. 3). We first evaluated $K_V2.1$-$K_V2.1$ proximity within isolated mesenteric smooth muscle cells by using two different antibodies directed against different epitopes in the $K_V2.1$ cytoplasmic C-terminus. In this case both intra- and inter-molecular proximity of the two epitopes would yield a PLA signal. Inter-molecular proximity could be visualized as "puncta", and we hypothesized that more puncta would be exhibited in cells where $K_V2.1$ was more clustered since there would be increased intermolecular proximity, allowing for more PLA reactions to occur. Confocal images of $K_V2.1_{WT}$ and $K_V2.1_{S590A}$ myocytes subjected to PLA show that puncta of $K_V2.1$-$K_V2.1$ PLA signals were randomly distributed throughout the cell, and that PLA signal could be detected in all cells consistent with our confocal data above showing that $K_V2.1$ microclustering still occurs in mutant myocytes of both sexes (Supplementary Fig. 3a). The density of PLA puncta of 0.035 ± 0.003 puncta/μm² (median = 0.035 μm²) in $K_V2.1_{WT}$ males was similar to 0.030 ± 0.003 puncta/μm² (median = 0.028 μm²) in $K_V2.1_{S590A}$ male mice ($P = 0.148$) (Supplementary Fig. 3b). Total puncta area per cell was similar between $K_V2.1_{WT}$ males at 8.494 ± 1.143 um² (median = 7.376 um²) and $K_V2.1_{S590A}$ males at 6.236 ± 2.166 um² (median = 2.3838 um²) ($P = 0.1578$) (Supplementary Fig. 3c). Consistent with the confocal imaging analysis, the density of $K_V2.1$-$K_V2.1$ PLA puncta was greater in $K_V2.1_{WT}$ females, with an average of 0.153 ± 0.011 puncta/μm² (median = 0.162 μm²), compared to 0.046 ± 0.003 puncta/μm² (median = 0.037 μm²) in $K_V2.1_{S590A}$ females ($P < 0.0001$) (Supplementary Fig. 3d). Similarly, total puncta area per cell was larger in $K_V2.1_{WT}$ females at 24.33 ± 2.289 um² (median = 23.90 um²) and $K_V2.1_{S590A}$ females at 6.484 ± 0.968 um² (median = 5.338 um²) ($P = < 0.0001$) (Supplementary Fig. 3e). Together, this data suggests that the $K_V2.1_{S590A}$ mutation reduces the level of $K_V2.1$ clustering in female myocytes.

Previous work from our group has shown that $K_V2.1$ expression promotes $Ca_V1.2$ clustering and activity in neurons[16] and arterial myocytes[15]. Following from this and the data above, we hypothesize that in arterial myocytes $K_V2.1$ plays a sex-specific structural role as an organizer to bring $Ca_V1.2$ channels together in female but not male myocytes. We again used PLA to test the hypothesis that the declustering of $K_V2.1$ channels in female but not male myocytes from $K_V2.1_{S590A}$ mice would decrease $K_V2.1$-$Ca_V1.2$ channel proximity in a sex-specific manner. Representative images of $K_V2.1$-$Ca_V1.2$ PLA puncta show randomly distributed interactions across the cell (Supplementary Fig. 3f).

Quantification showed that $K_V2.1$-$Ca_V1.2$ puncta density was unchanged between $K_V2.1_{WT}$ male myocytes with a mean of 0.022 ± 0.002 puncta/μm² (median = 0.016 puncta/μm²) and $K_V2.1_{S590A}$ male myocytes with a mean of 0.018 ± 0.002 puncta/μm² (median = 0.016 puncta/μm²; $P = 0.217$) (Supplementary Fig. 3g).

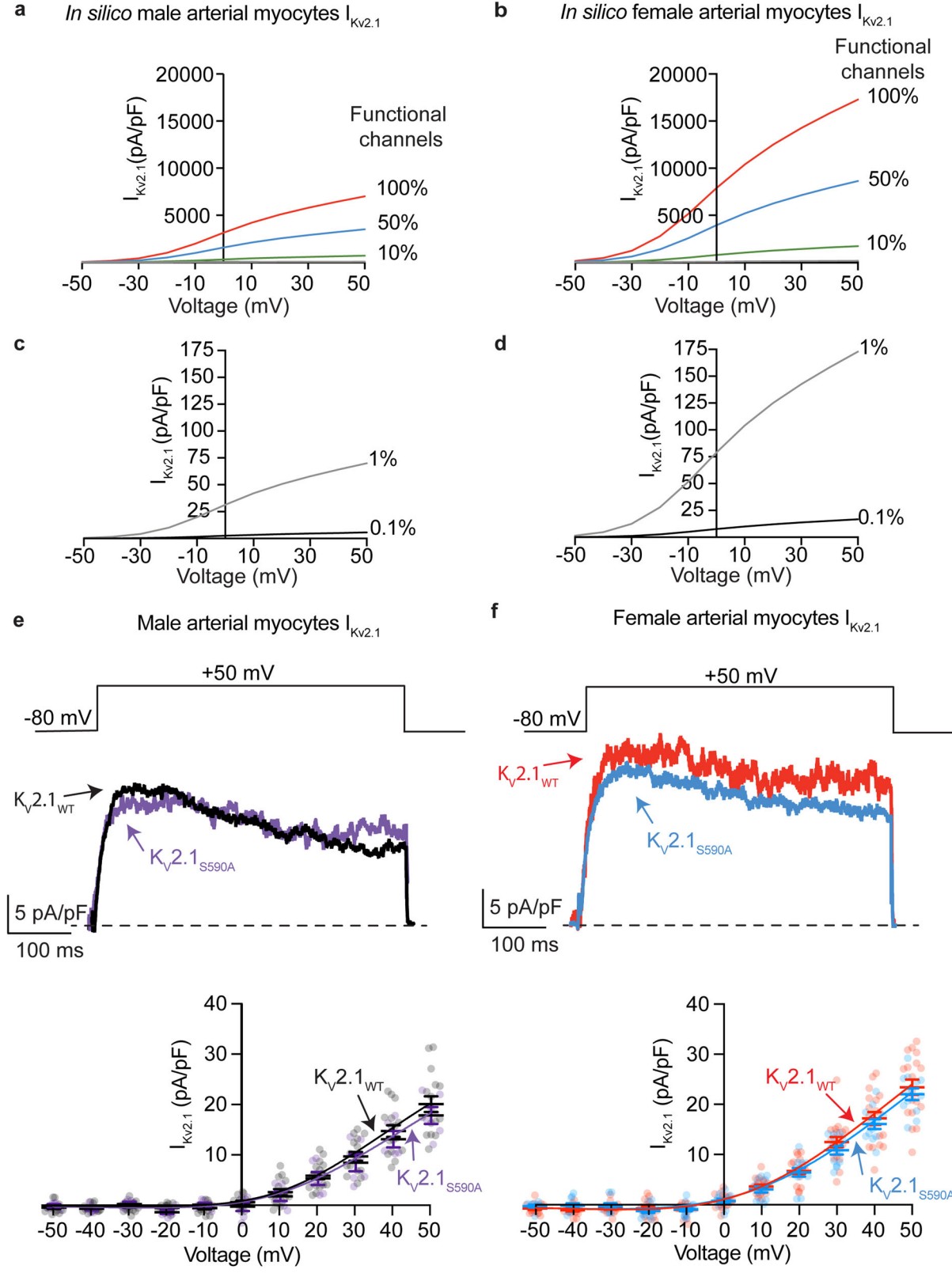

Total puncta area per cell was similar between $K_V2.1_{WT}$ males at $4.443 \pm 0.5655$ um$^2$ (median $= 2.700$ um$^2$) and $K_V2.1_{S590A}$ males at $4.398 \pm 0.4332$ um$^2$ (median $= 3.476$ um$^2$) ($P = 0.4746$) (Supplementary Fig. 3h). However, $K_V2.1$-$Ca_V1.2$ proximity decreased in female $K_V2.1_{S590A}$ myocytes with a mean of $0.030 \pm 0.004$ puncta/$\mu m^2$ (median $= 0.026$ puncta/$\mu m^2$) compared to $K_V2.1_{WT}$ female myocytes with a mean of $0.044 \pm 0.004$ puncta/$\mu m^2$ (median $= 0.026$ puncta/$\mu m^2$; $P = 0.013$) (Supplementary Fig. 3i). Total puncta area per cell was larger in $K_V2.1_{WT}$ females at $6.328 \pm 0.7606$ um$^2$ (median $= 4.171$ um$^2$) than $K_V2.1_{S590A}$ females at $4.268 \pm 0.5517$ um$^2$ (median $= 3.504$ um$^2$) ($P = 0.0217$) (Supplementary Fig. 3j). These data further support a sex-specific structural role for $K_V2.1$ channels, facilitating $Ca_V1.2$-$Ca_V1.2$ clustering.

**Fig. 4 Expression of clustering impaired K$_V$2.1$_{S590A}$ does not affect K$_V$2.1 channel activity in arterial myocytes.** Computationally modeled I$_{Kv2.1}$ in male (**a**) and female (**b**) myocytes assuming 100% (red), 50% (blue) or 10% (green) of K$_V$2.1 channels present in the plasma membrane are functional. Computationally modeled I$_{Kv2.1}$ in male (**c**) and female (**d**) myocytes assuming 1% (gray) and 0.1% (black) K$_V$2.1 channels are functional. **e** Representative I$_{Kv2.1}$ traces at +50 mV from K$_V$2.1$_{WT}$ male (black) and K$_V$2.1$_{S590A}$ male (purple) arterial myocytes and voltage dependence of I$_{Kv2.1}$ in K$_V$2.1$_{WT}$ (black) and K$_V$2.1$_{S590A}$ (purple) male myocytes from −50 to +50 mV. **f** Representative I$_{Kv2.1}$ traces at +50 mV from K$_V$2.1$_{WT}$ female (red) and K$_V$2.1$_{S590A}$ (blue) arterial myocytes and voltage dependence of I$_{Kv2.1}$ in K$_V$2.1$_{WT}$ (red) and K$_V$2.1$_{S590A}$ (blue) female myocytes from −50 to +50 mV. I$_{Kv2.1}$ traces were obtained by subtracting currents after the application of RY785 from control I$_K$ traces. $n = 18$ K$_V$2.1$_{WT}$ male, 6 K$_V$2.1$_{S590A}$ male, 19 K$_V$2.1$_{WT}$ female and 13 K$_V$2.1$_{S590A}$ myocytes. Error bars indicate mean ± SEM.

**Female myocytes expressing K$_V$2.1$_{S590A}$ have reduced macroscopic Ca$_V$1.2 currents.** We recorded macroscopic Ca$_V$1.2 currents (I$_{Ca}$) from male and female K$_V$2.1$_{WT}$ and K$_V$2.1$_{S590A}$ arterial myocytes (Fig. 5a–d). I$_{Ca}$ was activated by applying 300 ms voltage step depolarizations from a holding potential of −80 to +60 mV. We show I$_{Ca}$ traces recorded during a depolarization to 0 mV from representative male (Fig. 5a) and female (Fig. 5b) K$_V$2.1$_{WT}$ and K$_V$2.1$_{S590A}$ arterial myocytes. Note that the amplitude and kinetics of I$_{Ca}$ in these male K$_V$2.1$_{WT}$ and K$_V$2.1$_{S590A}$ arterial myocytes were similar. By contrast, we found that peak I$_{Ca}$ was smaller in female Kv2.1$_{S590A}$ myocytes compared to those in K$_V$2.1$_{WT}$ cells. In Fig. 5c and d, we show the voltage dependence of the amplitude of I$_{Ca}$ from all the cells examined over a wider range of membrane potentials. This analysis shows that the amplitude of I$_{Ca}$ is similar in K$_V$2.1$_{WT}$ and K$_V$2.1$_{S590A}$ male myocytes at all voltages examined. However, in female myocytes, I$_{Ca}$ was smaller in K$_V$2.1$_{S590A}$ than in K$_V$2.1$_{WT}$ at all voltages examined. Indeed, at 0 mV, I$_{Ca}$ amplitude in K$_V$2.1$_{S590A}$ cells was approximately 50% of that of WT females.

Next, we determined the level of expression of Ca$_V$1.2 protein in male and female K$_V$2.1$_{S590A}$ and K$_V$2.1$_{WT}$ vessels using RT-PCR and immunocytochemistry (Fig. 5e, f) approaches. Our analysis suggests that mRNA expression (Fig. 5g) and total Ca$_V$1.2 protein (Fig. 5h) is similar in K$_V$2.1$_{S590A}$ and K$_V$2.1$_{WT}$ males. Likewise, mRNA expression (Fig. 5i) and total Ca$_V$1.2 protein (Fig. 5j) is similar in female K$_V$2.1$_{S590A}$ and K$_V$2.1$_{WT}$ vessels. This suggests that the smaller I$_{Ca}$ in female K$_V$2.1$_{S590A}$ than K$_V$2.1$_{WT}$ myocytes is not likely due to lower Ca$_V$1.2 expression in these cells.

**K$_V$2.1$_{S590A}$ mutation decreases Ca$_V$1.2 cluster sizes in female but not male arterial myocytes.** In a previous study[15], we suggested a model that differences in I$_{Ca}$ amplitude between female and male arterial myocytes were due to sex-specific differences in K$_V$2.1-mediated Ca$_V$1.2 clustering that impacted the probability of cooperative gating of these channels. Our data above show differences in I$_{Ca}$ amplitude between female K$_V$2.1$_{WT}$ and K$_V$2.1$_{S590A}$ myocytes in the absence of differences in Ca$_V$1.2 expression levels. Thus, we investigated whether K$_V$2.1$_{S590A}$ expression altered Ca$_V$1.2 channel clustering in a sex-specific manner using GSD super-resolution microscopy (Fig. 6).

We show GSD run in TIRF mode super-resolution images from representative male (Fig. 6a) myocytes from K$_V$2.1$_{WT}$ and K$_V$2.1$_{S590A}$ mice. The insets show expanded views of two regions of interest (1 μm$^2$) within each cell image. Our TIRF images show that Ca$_V$1.2 clusters of various sizes are expressed throughout these cells. The frequency distribution of Ca$_V$1.2 cluster areas in K$_V$2.1$_{WT}$ and K$_V$2.1$_{S586A}$ male could both be fit with an exponential function (Fig. 6b). The mean area of Ca$_V$1.2 clusters in male K$_V$2.1$_{WT}$ of 2259 ± 55 nm$^2$ (median = 2219 nm$^2$) was similar to the K$_V$2.1$_{S590A}$ male mean of 2345 ± 82 nm$^2$ (median = 2354 nm$^2$) ($P = 0.173$) (Fig. 6c), suggesting that S590A mutation in male myocytes does not affect Ca$_V$1.2 channel clustering.

Figure 6d shows representative super-resolution images from female myocytes of K$_V$2.1$_{WT}$ and K$_V$2.1$_{S590A}$ mice. Similar to males, the frequency distribution of Ca$_V$1.2 cluster areas in K$_V$2.1$_{WT}$ and K$_V$2.1$_{S586A}$ females could both be fit with an exponential function (Fig. 6e). However, Ca$_V$1.2 cluster sizes were significantly smaller in K$_V$2.1$_{S590A}$ female myocytes with a mean area of 2381 ± 91 nm$^2$ (median = 2251 nm$^2$) compared to K$_V$2.1$_{WT}$ female myocytes whose mean area was 3098 ± 164 nm$^2$ (median = 3117 nm$^2$) ($P = 0.0001$) (Fig. 6f). Taken together with our electrophysiological data, our findings suggest that the clustering and activity of Ca$_V$1.2 channels is modulated by the degree of K$_V$2.1 clustering.

As shown in Supplementary Fig. 4a, b, our stochastic self-assembly model effectively reproduced the steady-state size distributions that we measured for Ca$_V$1.2 clustering in K$_V$2.1$_{WT}$ and K$_V$2.1$_{S586A}$ arterial myocytes. The parameters used in the model are summarized in Supplementary Fig. 4c. These in silico data suggest that Ca$_V$1.2 clusters in K$_V$2.1$_{S590A}$ female myocytes have a higher probability of growth than those in female K$_V$2.1$_{WT}$ arterial myocytes.

**K$_V$2.1$_{S586A}$ reduces Ca$_V$1.2-Ca$_V$1.2 channel interactions.** Having previously shown that K$_V$2.1 enhances Ca$_V$1.2-Ca$_V$1.2 channel interactions in arterial myocytes[15], we used a split-Venus fluorescent protein system to determine if a decrease in K$_V$2.1 macro-clustering would lead to a reduction in Ca$_V$1.2-Ca$_V$1.2 interactions and channel activity. This system involves fusing Ca$_V$1.2 channels with either the N-terminal fragment (Ca$_V$1.2-VN) or the C-terminal fragment (Ca$_V$1.2-VC) of Venus protein. Individually, neither Ca$_V$1.2-VN nor Ca$_V$1.2-VC emits fluorescence. However, when brought into close enough proximity for interaction to occur, they can reconstitute a full fluorescence emitting Venus protein. Thus, the split-Venus fluorescence can be utilized to indicate interactions between neighboring Ca$_V$1.2 channels. Accordingly, we compared the split-Venus fluorescence in HEK293T cells expressing Ca$_V$1.2-VN and Ca$_V$1.2-VC and co-expressing either K$_V$2.1$_{WT}$ or K$_V$2.1$_{S586A}$ (Fig. 7). The voltage protocols used for these experiments to promote Venus reconstitution via the complementary interaction of clustered Ca$_V$1.2-VN and Ca$_V$1.2-VC are similar to those used in two recent studies[10,15] and are described in detail in the Methods section of this paper. Briefly, we recorded I$_{Ca}$ in response to short 200 ms depolarization steps to voltages ranging from −50 to +50 mV before and after the application of a conditioning 9 s step depolarization to 0 mV, to allow for Venus reconstitution between adjacent Ca$_V$1.2-VN and Ca$_V$1.2-VC channels.

We first transfected HEK293T cells with Ca$_V$1.2-VN, Ca$_V$1.2-VC, and the non-conducting but macro-clustering-competent rat K$_V$2.1$_{P404W}$ channel[37] tagged with red-shifted fluorescent protein DsRed. The P404W mutation confers a non-conductive K$_V$2.1 phenotype, allowing us to study the structural clustering role of K$_V$2.1 without masking of the Ca$^{2+}$ currents by K$^+$. Our data show that submitting these cells to the conditioning protocol increased I$_{Ca}$ at most membrane potentials, with the peak I$_{Ca}$ at

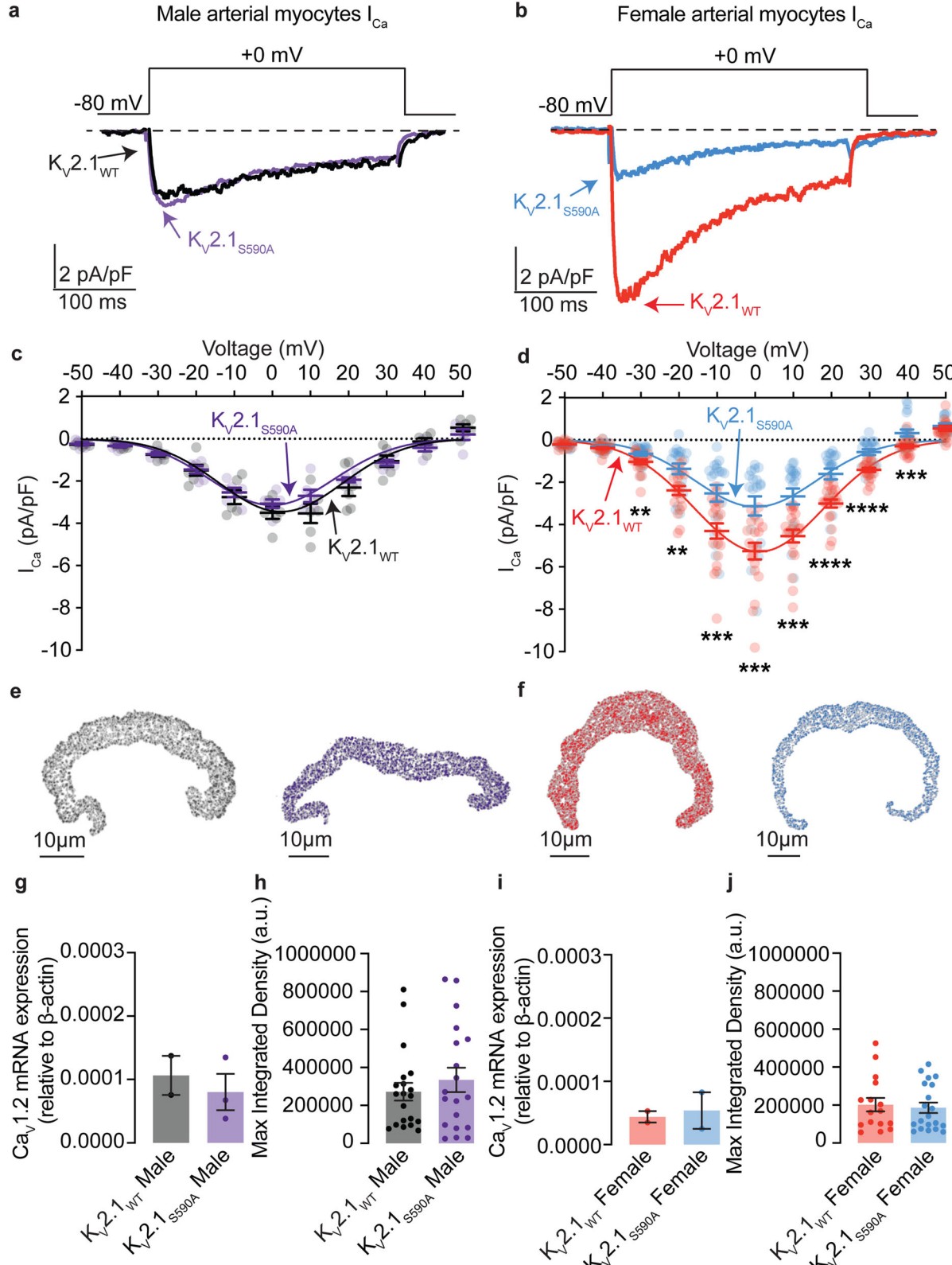

0 mV being about 51% larger compared to control (i.e., before application of conditioning pulse) currents (Fig. 7a).

We next tested the hypothesis that reduction of $K_V2.1$ macro-clusters with the S586A point mutation would decrease $Ca_V1.2$ channel activity. We co-expressed $Ca_V1.2$-VN, $Ca_V1.2$-VC and DsRed-$K_V2.1_{P404W,S586A}$ in HEK293T cells and repeated the above protocol. We found that $I_{Ca}$ exhibited an increase of about

9% between pre- and post- conditional protocols at 0 mV (Fig. 7b).

Analysis of the associated TIRF images show that Venus fluorescence increased by approximately 21% with stimulation from control to post-conditioning steps in cells cotransfected with $Ca_V1.2$-VN, $Ca_V1.2$-VC and DsRed-$K_V2.1_{P404W}$, suggesting an increase in $K_V2.1$ macro-cluster dependent $Ca_V1.2$-$Ca_V1.2$

**Fig. 5 $I_{Ca}$ is reduced in $K_V2.1_{S590A}$ female myocytes but unaffected in male arterial myocytes.** $I_{Ca}$ records (0 mV) from representative $K_V2.1_{WT}$ and $K_V2.1_{S590A}$ male (**a**) and $K_V2.1_{WT}$ and $K_V2.1_{S590A}$ female (**b**) myocytes. Voltage dependence of $I_{Ca}$ from male (**c**) and female (**d**) myocytes at membrane potentials ranging from −50 to +50 mV ($n = 6$ $K_V2.1_{WT}$ male, 7 $K_V2.1_{S590A}$ male, 20 $K_V2.1_{WT}$ female and 20 $K_V2.1_{S590A}$ myocytes). Representative images of immunolabeled Ca$_V$1.2 in myocytes from $K_V2.1_{WT}$ male (**e**, black), $K_V2.1_{S590A}$ male (**e**, purple), $K_V2.1_{WT}$ female (**f**, red), and $K_V2.1_{S590A}$ female (**f**, blue) mice. **g** Summary data from real-time quantitative PCR experiments of Ca$_V$1.2 mRNA expression relative to β-actin in male ($n = 2$ from 4 $K_V2.1_{WT}$ and 4 $K_V2.1_{S590A}$ males where 2 artery beds were pooled for each sample). **h** Quantification of immunofluorescence of labeled Ca$_V$1.2α subunits in male myocytes ($n = 20$ $K_V2.1_{WT}$ and 19 $K_V2.1_{S590A}$ male myocytes). **i** Summary data from real-time quantitative PCR experiments of Ca$_V$1.2 mRNA expression relative to β-actin in female ($n = 2$ from 4 $K_V2.1_{WT}$ and 4 $K_V2.1_{S590A}$ females where 2 artery beds were pooled for each sample). **j** Quantification of immunofluorescence of labeled Ca$_V$1.2α subunits in female myocytes ($n = 16$ $K_V2.1_{WT}$ and 20 $K_V2.1_{S590A}$ myocytes). *$P < 0.05$, **$P < 0.01$, ***$P < 0.001$, ****$P < 0.0001$. Error bars indicate mean ± SEM.

interactions (Fig. 7c, d). Furthermore, Venus fluorescence with $K_V2.1_{P404W,S586A}$ expression increased by about 9%, a level similar to what was previously published[15,17] with Ca$_V$1.2-VN and Ca$_V$1.2 VC alone (Fig. 7d). We propose this increase is due to the intrinsic ability of Ca$_V$1.2 channels to interact with one another. Together these data further support a model for the structural role $K_V2.1$ macro-clusters play in enhancing Ca$_V$1.2-Ca$_V$1.2 interactions and activity.

**The activity of Ca$_V$1.2 channels is reduced in $K_V2.1_{S590A}$ female but not male arterial myocytes.** We next examined whether variations in the activity of Ca$_V$1.2 channels could explain the differences in $I_{Ca}$ observed in myocytes from $K_V2.1_{WT}$ and $K_V2.1_{S590A}$ male and female mice. Ca$_V$1.2 channel activity was determined by recording Ca$_V$1.2 sparklets using TIRF microscopy as previously described[6,7,12,17,38–40] (Fig. 8). TIRF microscopy of near-plasma membrane intracellular Ca$^{2+}$ levels provides a powerful tool for recording Ca$^{2+}$ entry via individual or small clusters of Ca$_V$1.2 channels, as it enables the activity of individual channels to be recorded from a relatively large membrane area allowing for the identification of discrete sarcolemma signaling domains. In this analysis, Ca$_V$1.2 sparklet activity is expressed as $nP_s$, where n is the number of quantal levels reached by the sparklet site and $P_s$ is the probability of sparklet occurrence. Detailed analysis of Ca$_V$1.2 sparklets sites[38] revealed heterogeneity in activity at different sites. Therefore, Ca$_V$1.2 sparklets sites were separated into low and high activity sites, using an $nP_s$ cutoff of 0.2.

Representative Ca$_V$1.2 sparklet traces are provided from low activity sparklet sites. Of note, the majority of the sparklet activity that occurs in male myocytes is produced by a signal that corresponds to a single channel opening (one quantal unit) (Fig. 8a, b). The strength of the coupled gating is denoted by the κ value, and in these traces, the κ values are close to or equal to 0, indicating no or weak coupling between the channels.

We found that in low activity sparklet sites, the average $nP_s$ was not different between $K_V2.1_{WT}$ and $K_V2.1_{S590A}$ male myocytes (Fig. 8c). $K_V2.1_{WT}$ sparklet sites had an average $nP_s$ of $0.07 \pm 0.01$ (median = 0.07) compared to $K_V2.1_{S590A}$ where the $nP_s$ average was $0.07 \pm 0.02$ (median = 0.06) ($P = 0.46$). In $K_V2.1_{WT}$ male myocytes, we detected 7 high activity sparklet sites with a mean $nP_s$ of $0.25 \pm 0.02$ (median = 0.24) (Fig. 8c). Interestingly, we could not detect any high activity sparklets sites from $K_V2.1_{S590A}$ male myocytes. We therefore included cells from where we could not record any sparklet activity from. Including this data, the average $nP_s$ of sparklets recorded from $K_V2.1_{WT}$ males were $0.13 \pm 0.04$ compared to an $nP_s$ of 0 from $K_V2.1_{S590A}$ males. Furthermore, we did not observe a difference in the number of Ca$_V$1.2 sparklet sites with an average of $1.50 \pm 0.17$ sites (median = 1.50 sites) in $K_V2.1_{WT}$ males compared with $1.2 \pm 0.20$ sites (median = 1.00 sites) in $K_V2.1_{S590A}$ male myocytes ($P = 0.15$) (Fig. 8d).

Previous work[40] showed that Ca$_V$1.2 sparklet sites appeared to arise from the simultaneous opening and/or closing of multiple channels suggesting that small groups of channels may be functioning cooperatively. To examine such coupling, we employed a coupled Markov chain model to determine the coupling coefficient (κ) among Ca$_V$1.2 channels at Ca$^{2+}$ sparklet sites. The κ value ranges from 0 for channels that gate independently to 1 for channels that are tightly coupled and open and close simultaneously. A detailed description of this model is provided in the expanded Methods section. Using this analysis, we found that the average κ value of $0.28 \pm 0.09$ (median = 0.30) in $K_V2.1_{WT}$ male myocytes was not significantly different from $0.20 \pm 0.09$ (median = 0.22) in $K_V2.1_{S590A}$ male myocytes ($P = 0.27$) (Fig. 8e).

In contrast, the $K_V2.1_{WT}$ female trace (Fig. 8f) from a low activity site exhibited coordinated multi-channel openings, of up to 3 channels with a κ value of 0.466. Interestingly, the activity of sparklet sites from $K_V2.1_{S590A}$ female myocytes (Fig. 8g) were similar to those of $K_V2.1_{WT}$ and $K_V2.1_{S590A}$ male myocytes, exhibiting mostly single channel openings and few coupled gating events.

When we compared $nP_s$ in low activity sites in $K_V2.1_{WT}$ and $K_V2.1_{S590A}$ female myocytes, we could not discern a difference in their average $nP_s$ values (Fig. 8h). $K_V2.1_{WT}$ female mean $nP_s$ was $0.10 \pm 0.01$ (median = 0.09), which was similar to $0.06 \pm 0.03$ (median = 0.04) in $K_V2.1_{S590A}$ female myocytes ($P = 0.08$). High activity sites averaged an $nP_s$ of $0.31 \pm 0.02$ (median = 0.30) in $K_V2.1_{WT}$ cells. We only recorded a single high activity sparklet site with an $nP_s$ of 0.29 in $K_V2.1_{S590A}$ female cells, similar to that of $K_V2.1_{WT}$ female myocytes ($P = 0.50$) (Fig. 8h). As with the male myocytes, we included cells with no sparklet activity where the average $nP_s$ in $K_V2.1_{WT}$ females was $0.20 \pm 0.04$ (median = 0.27) compared to $0.05 \pm 0.05$ (median = 0.0) in $K_V2.1_{S590A}$ female myocytes. Additionally, $K_V2.1_{WT}$ female myocytes exhibited $2.73 \pm 0.27$ (median = 3.0) Ca$_V$1.2 sparklet sites per cell, higher than $1.40 \pm 0.25$ (median = 1.0) Ca$_V$1.2 sparklet sites in $K_V2.1_{S590A}$ female myocytes ($P = 0.008$) suggesting decreased Ca$_V$1.2 channel activity with S590A mutation (Fig. 8i). Together this data suggests a cumulative higher activity in $K_V2.1_{WT}$ female myocytes.

However, the average κ value of $0.36 \pm 0.05$ (median = 0.38) in female $K_V2.1_{WT}$ myocytes, was significantly higher than $0.14 \pm 0.07$ (median = 0) ($P = 0.0082$) in female $K_V2.1_{S590A}$ myocytes, suggesting more coupled events (Fig. 8j). Taken together, these data indicate increased Ca$_V$1.2 channel activity and coupled gating in myocytes from $K_V2.1_{WT}$ females compared to those with the $K_V2.1_{S590A}$ mutation, suggesting that clustering of $K_V2.1$ modulates Ca$_V$1.2 channel activity.

**Discussion**

In this study, we show that arterial smooth muscle cells from mice expressing a gene-edited point mutation of the $K_V2.1$ channel that selectively eliminates its characteristic macro-clustered

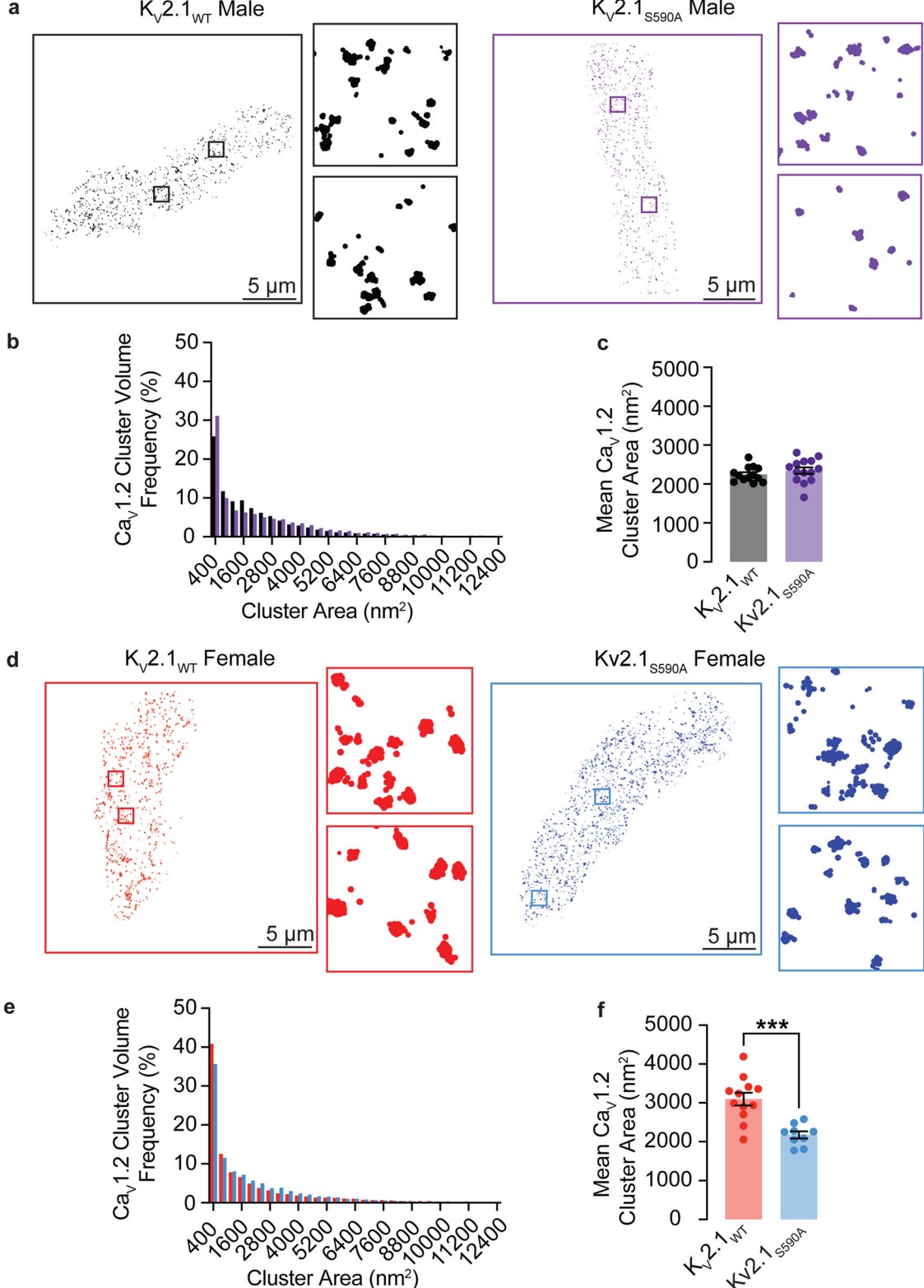

**Fig. 6 Ca$_V$1.2 cluster sizes are decreased in myocytes from female but not male K$_V$2.1$_{S590A}$ mice. a** Representative super-resolution GSD images of immunolabeled Ca$_V$1.2 labeled channels in K$_V$2.1$_{WT}$ and K$_V$2.1$_{S590A}$ male myocytes. Insets show 4 µm$^2$ regions of interest. **b** Relative frequency as a percentage of Ca$_V$1.2 cluster areas of K$_V$2.1$_{WT}$ (black) and K$_V$2.1$_{S590A}$ (purple) in male myocytes. **c** Summary plot of mean Ca$_V$1.2 cluster areas in male myocytes ($n = 13$ K$_V$2.1$_{WT}$ and 14 K$_V$2.1$_{S590A}$ male myocytes). **d** Representative super-resolution GSD microscopy images of immunolabeled Ca$_V$1.2 labeled channels in K$_V$2.1$_{WT}$ and K$_V$2.1$_{S590A}$ female myocytes. **e** Relative frequency as a percentage of Ca$_V$1.2 cluster areas of K$_V$2.1$_{WT}$ (red) and K$_V$2.1$_{S590A}$ (blue) female myocytes. **f** Summary plot of mean Ca$_V$1.2 cluster areas in female myocytes ($n = 12$ K$_V$2.1$_{WT}$ and 9 K$_V$2.1$_{S590A}$ female myocytes). *$P < 0.05$, **$P < 0.01$, ***$P < 0.001$, ****$P < 0.0001$. Error bars indicate mean ± SEM.

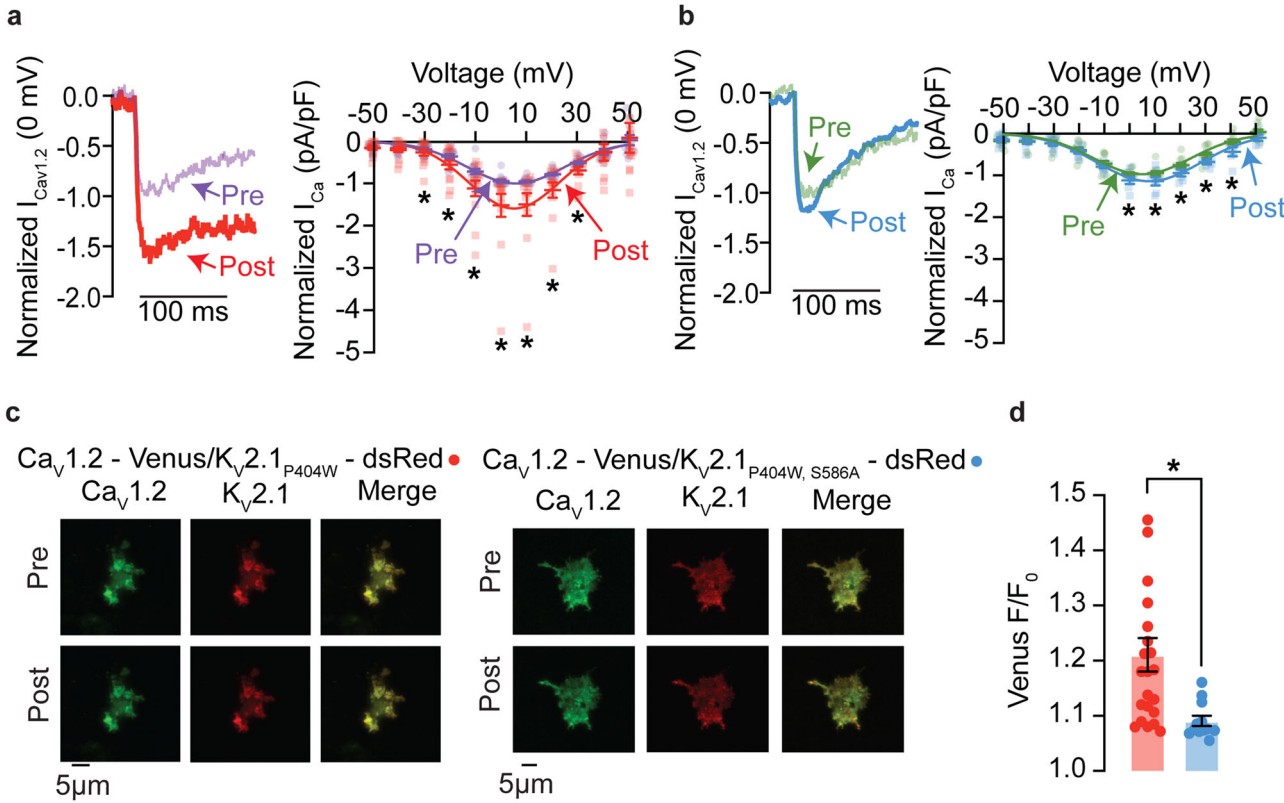

**Fig. 7 $Ca_V1.2$-$Ca_V1.2$ interactions are decreased in cells expressing $K_V2.1_{S586A}$. a** Representative currents measured at 0 mV from pre- (purple) and post- (red) conditioning protocol from HEK293T cells expressing $Ca_V1.2$-VN, $Ca_V1.2$-VC, and DsRed-$K_V2.1_{P404W}$. Normalized (to peak current in pre-conditioning protocol) pre- and post- IV relationships from HEK293T cells expressing $Ca_V1.2$-VN, $Ca_V1.2$-VC, and DsRed-$K_V2.1_{P404W}$. **b** Representative currents measured at 0 mV from pre- (green) and post- (blue) conditioning protocol from HEK293T cells expressing $Ca_V1.2$-VN, $Ca_V1.2$-VC, and DsRed-$K_V2.1_{P404W,S586A}$. Normalized pre- and post- IV relationships from HEK293T cells expressing $Ca_V1.2$-VN, $Ca_V1.2$-VC, and DsRed-$K_V2.1_{P404W,S586A}$ ($n = 13$ DsRed-$K_V2.1_{P404W}$ and 10 DsRed-$K_V2.1_{P404W,S586A}$ cells). **c** Representative TIRF images of Venus fluorescence reconstitution in HEK293T cells from cells transfected with $Ca_V1.2$-VN, $Ca_V1.2$-VC and DsRed-$K_V2.1_{P404W}$ or $Ca_V1.2$-VN, $Ca_V1.2$-VC and DsRed-$K_V2.1_{P404W,S586A}$. Pre- and post-conditioning $Ca_V1.2$-Venus (green), $K_V2.1_{P404W}$ or $K_V2.1_{P404W,S856A}$ (red), and the merge of the two channels are presented. **d** Summary of $Ca_V1.2$-Venus fluorescence (F/$F_0$) ($n = 22$ DsRed-$K_V2.1_{P404W}$ and 12 DsRed-$K_V2.1_{P404W,S586A}$ cells). *$P < 0.05$. Error bars indicate mean ± SEM.

localization have properties remarkably like those from $K_V2.1$ knock-out mice. This leads us to formulate a new model in which $K_V2.1$ expression, by itself, is not sufficient for this channel to exert its structural functions on modulating $Ca_V1.2$ clustering and activity, but rather depends on $K_V2.1$ channel's capacity to form macro-clusters. Notably, the presence of $K_V2.1$ macro-clusters in female, but not male myocytes underlie sex-specific differences in $Ca^{2+}$ influx via $Ca_V1.2$ channels in arterial smooth muscle (Fig. 9). Our data suggest a new paradigm whereby the clustering of ion channels underlies their physiological functions, independent of their ability to conduct ions.

Analysis of super-resolution images indicates that clustering of $K_V2.1$ and $Ca_V1.2$ channels is random and hence does not involve an active process. This stochastic self-assembly mechanism leads to micro- and macro-clusters of varying sizes that represent the default organization of $K_V2.1$ and $Ca_V1$ channels expressed endogenously in neurons and smooth muscle cells or exogenously in heterologous cells[9]. Furthermore, we found that $K_V2.1$ macro-clusters are composed of groups of micro-clusters. This is consistent with a recent study showing that in developing neurons $K_V2.1$ macro-clusters formed from the coalescence of numerous micro-clusters[41] and suggests that the organization of $K_V2.1$ clusters is hierarchical.

An important finding in this study is that $K_V2.1$ clustering is more prominent in female than in male arterial myocytes, with female myocytes expressing a larger proportion of macro-clusters.

In this context, the development of the $K_V2.1_{S590A}$ mouse allowed us to investigate the separatable structural clustering and ion conducting roles of this channel. We found that expression $K_V2.1_{S590A}$ nearly eliminated macro-clustering in female myo-cytes but had no impact on $K_V2.1$ micro-clusters in cells from male or female myocytes. Because the S590A mutation eliminated a phosphorylation site in the PRC domain that causes macro-clustering, these findings suggested that the potential mechanism of these sex-specific differences in $K_V2.1$ clustering was differential phosphorylation of this specific serine in male and female myocytes.

Indeed, $K_V2.1$ phosphorylation and macro-clustering is regulated by a myriad of protein kinases such as CDK5 and protein phosphatases such as calcineurin[42]. These kinases and phosphatases work as a rheostatic mechanism to regulate the phosphorylation status of $K_V2.1$ based on physiological demands[26,27,42]. Accordingly, we found that that the phosphorylation state of $K_V2.1$ in arterial myocytes differs between the two sexes, specifically, that $K_V2.1$ in male myocytes is phosphorylated to a much lower degree.

It is intriguing to consider the potential clustering mechanisms that are impacted by inhibiting phosphorylation at the 590/586 specific site of the PRC domain by the serine to alanine point mutation. One hypothesis is that VAP proteins act to modulate the probability of macro-cluster formation. Studies show that $K_V2.1$ clusters are expressed at sites where the endo/sarcoplasmic

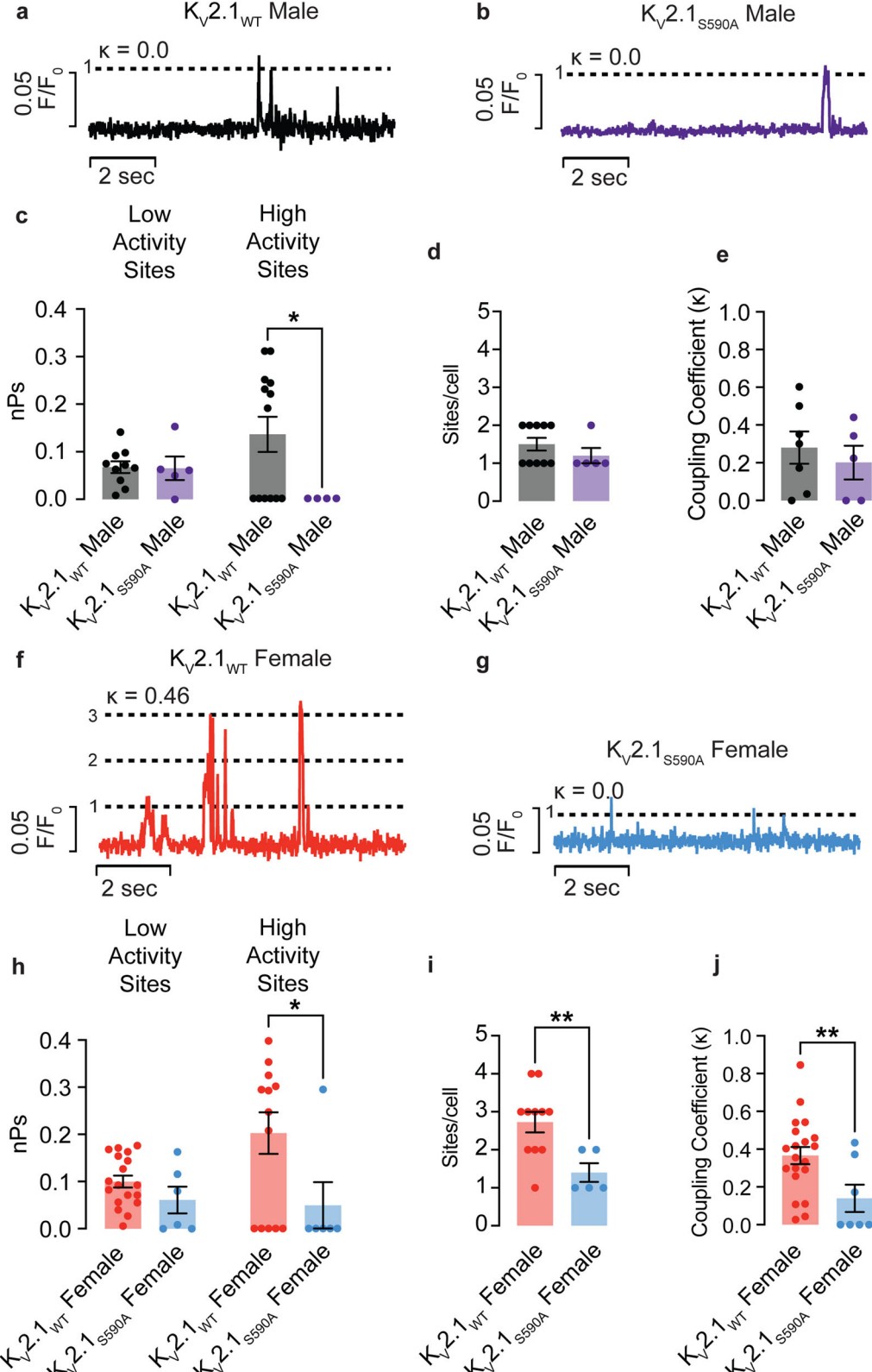

**Fig. 8 Activity of Ca$_V$1.2 channels is reduced in K$_V$2.1$_{S590A}$ female but not male arterial myocytes.** Representative sparklet traces from K$_V$2.1$_{WT}$ male (**a**) and K$_V$2.1$_{S590A}$ male (**b**) myocytes. κ values are shown above each trace. **c** nPs values from low and high activity sites in male myocytes. **d** Coupling coefficient values (κ) from male myocytes. **e** Sparklet sites per cell from male myocytes ($n = 16$ K$_V$2.1$_{WT}$ and 9 K$_V$2.1$_{S590A}$ male myocytes). Representative sparklet traces from K$_V$2.1$_{WT}$ female (**f**) and K$_V$2.1$_{S590A}$ female (**g**) myocytes. **h** nPs values from low and high activity sites in female myocytes. **i** Coupling coefficient values (κ) from female myocytes. **j** Sparklet sites per cell from female myocytes ($n = 16$ K$_V$2.1$_{WT}$ and 10 K$_V$2.1$_{S590A}$ female myocytes. *$P < 0.05$, **$P < 0.01$. Error bars indicate mean ± SEM.

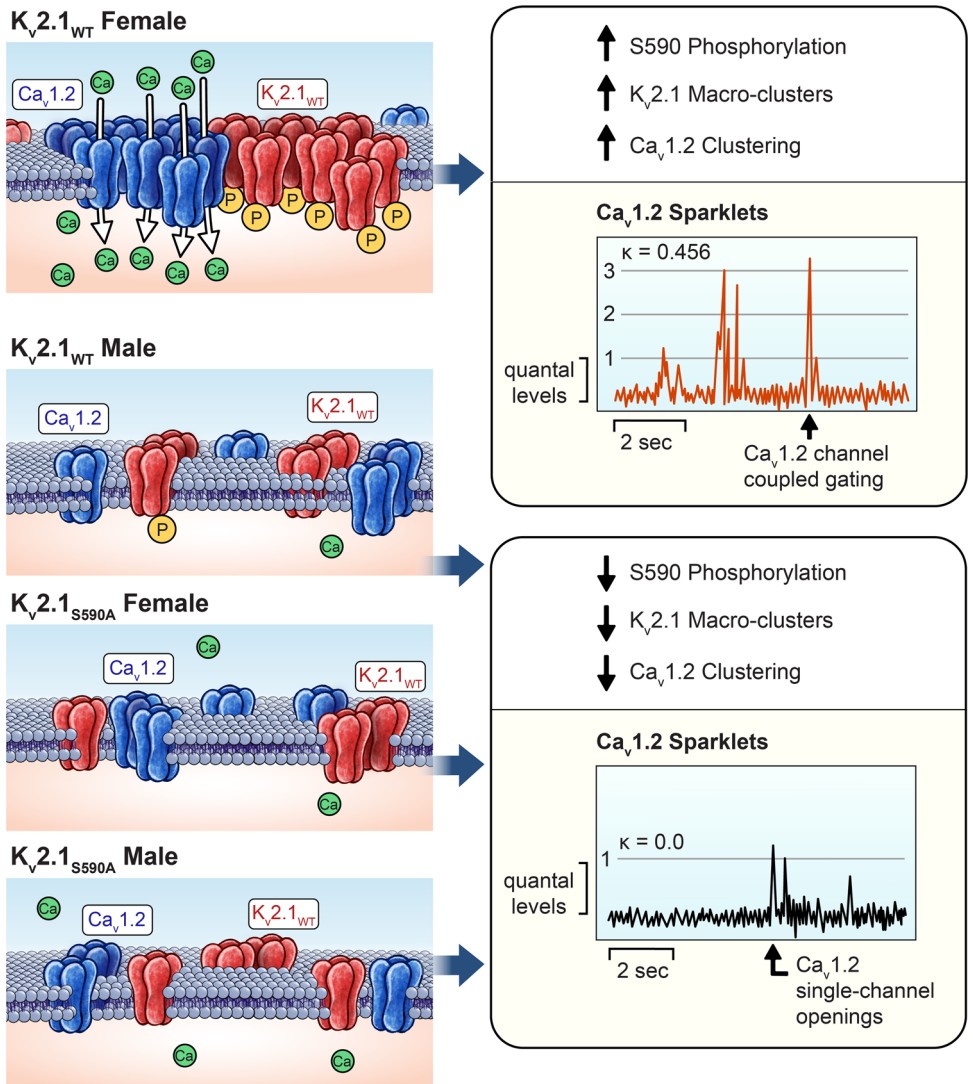

**Fig. 9 Proposed model by which $K_V2.1$ macro-clusters disruption eliminates sex-specific differences in $Ca^{2+}$ influx via $Ca_V1.2$ channels.** Enhanced S590A phosphorylation (yellow circles labeled with a "P") of $K_V2.1$ channels (red) in $K_V2.1_{WT}$ female myocytes increases $K_V2.1$ macro-clustering concomitant with larger $Ca_V1.2$ channel clusters (blue) and increased $Ca_V1.2$ channel activity (red trace). $K_V2.1_{WT}$ male and S590A mutant myocytes exhibit decreased or no S590 phosphorylation compared to $K_V2.1_{WT}$ female myocytes, disrupting the formation of $K_V2.1$ macro-clusters. $Ca_V1.2$ channel clusters are smaller and exhibit less activity.

reticulum is brought into close juxtaposition to the plasma membrane[41] and this interaction and accumulation of channels relies on the tethering of $K_V2.1$ to VAP proteins[21,43]. The transmembrane endo/sarcoplasmic reticulum VAP proteins (VAPA and VAPB) interact with the phosphorylated $K_V2.1$ PRC domain and have been proposed to function to increase the local concentration of $K_V2.1$ channels at endo/sarcoplasmic reticulum-plasma membrane junctions resulting in $K_V2.1$ macro-clustering.

Consistent with this, Kirmiz et al. [21], found that knock-out of VAPA in RAW664.7 macrophage cells resulted in a decrease in $K_V2.1$ channel clustering. Knockdown of endogenous VAP proteins similarly impaired clustering of $K_V2.1$ heterologously expressed in HEK293T cells[43]. Interestingly, the model proposed in these prior papers[21,43] suggested that the phosphorylated PRC domain is necessary and sufficient for macro-clustering of $K_V2$ channels. This is consistent with prior studies showing that mutations disrupting or eliminating the PRC domain[22,37,43] or treatments that impact $K_V2.1$ phosphorylation[26,27,42] impact $K_V2.1$ clustering. It is presumed that the phosphorylation of multiple serine residues, including S590, within the PRC domain

provide the negative charges needed to generate a functional VAP-binding FFAT — two phenylalanines in an acidic tract — motif, as has been shown for numerous other proteins that exhibit phosphorylation-dependent binding to VAPs[44]. Therefore, one possible mechanism for the decrease in macro-clustering in the S590A mutant is the inability of VAP proteins to recognize the PRC domain of mutated channels preventing cluster growth. The similarity in the patterns of cluster sizes and densities between HEK293T cells and arterial myocytes of both WT and S590A channels is noteworthy, indicating the possibility of a shared set of mechanisms. Further research will be necessary to uncover the underlying factors that govern these clustering patterns.

Prior studies have suggested that the bulk of $K_V2.1$ channels heterologously expressed in Xenopus oocytes[28] or HEK293T cells[19] as well as endogenous $K_V2.1$ in hippocampal neurons[45] and arterial myocytes[15] are in a nonconducting state. The prevailing view is that aggregation of $K_V2.1$ channels into high density clusters is what renders most of these channels incapable of conducting $K^+$[18]. Although our study does not

address this issue comprehensively, at a minimum, our data suggest that $K_V2.1$ conduction is not dependent on macro-clustering formation. Future studies should investigate whether the formation of $K_V2.1$ micro-clusters may be sufficient to electrically silence these channels.

We demonstrate the structural role of $K_V2.1$ clustering in regulating $Ca_V1.2$ channel clustering and activity that occurs in native cells. This is important because the generally accepted view is that the functional impact of ion channel clustering is to exclusively concentrate ion conducting roles at specific sites. For example, $Na^+$ channel clustering at nodes of Ranvier[46], neuronal $Ca^{2+}$ channel clustering at active zones in presynaptic terminals[47], and skeletal muscle $Ca^{2+}$ channels at SR $Ca^{2+}$ release units[48]. In the case of ventricular myocytes, it is concentrating voltage sensors at specific sites in the junctional dyad[49,50]. We propose that $K_V2.1$ clustering is distinct in playing a role in modulating the localization and activity of an otherwise seemingly unrelated ion channel: $Ca_V1.2$ channels. This functional impact of $K_V2.1$ is due to the density-dependent cooperative gating that is an intrinsic property of $Ca_V1.2$ channels[51].

The observation that the number of $K_V2.1$-$K_V2.1$ PLA puncta decreased with the S590A mutation suggests that the distance between individual channels increased above the PLA limits (i.e., ~50 nm[35]), which we propose contributes to a large loss of macro-clusters in cells expressing these clustering-deficient channels. However, the observation of a significant, albeit more modest decrease in $Ca_V1.2$-$K_V2.1$ puncta (i.e., compared to $K_V2.1$-$K_V2.1$ puncta) suggests to us that, on average, at least some $K_V2.1_{S590A}$ channels are still expressed within ~50 nm of a $Ca_V1.2$ cluster. The increase in the variance of the number of $Ca_V1.2$-$K_V2.1$ PLA puncta per cell due to the S590A mutation likely indicates increased independence, or randomness, in the spatial arrangement of the stochastically self-assembled $Ca_V1.2$ and $K_V2.1$ clusters.

Remarkably, the overall impact of $K_V2.1_{S590A}$ expression is that the differences between the $I_{Ca}$ amplitude of wild-type male and female myocytes were eliminated in myocytes expressing the $K_V2.1_{S590A}$ mutation, similar to what we observed in homozygous $K_V2.1$ knockout mice[15]. Thus, declustering $K_V2.1$ macro-clusters appears to have the same impact as fully eliminating $K_V2.1$ expression on $Ca_V1.2$ clustering and activity in male and female myocytes. As our work also suggests that in arterial myocytes the conductive function of $K_V2.1$ channels is independent of the degree of its clustering, in our model it is the extent of $K_V2.1$ clustering that is the key determinant of the sex-specific differences in $Ca^{2+}$ influx observed in these cells.

Our previous work established that $K_V2.1$ serves dual roles, encompassing both conductive and structural functions, which yield opposing functional consequences in arterial myocytes, with the former predominating in males and the latter in females[15]. In female myocytes, which exhibit greater expression of $K_V2.1$ protein than males, $K_V2.1$ has a larger structural role, serving as sites for larger clusters of $Ca_V1.2$, ultimately leading to higher $[Ca^{2+}]_i$, and greater myogenic tone than male myocytes. Together this would suggest that $K_V2.1$-$Ca_V1.2$ interactions contribute to the sex-specific regulation of vascular smooth muscle excitability, $Ca^{2+}$ dynamics and myogenic tone. Additional questions raised in this study regard the physiological consequences of the $K_V2.1_{S590A}$ mutation. We would hypothesize that $K_V2.1_{S590A}$ females would have decreased myogenic due to decreased $Ca_V1.2$ cluster size and activity. Future studies will aim to further elucidate the functional consequences of this mutation and whether targeting this interaction could provide a therapeutic option for blood pressure regulation.

The findings in this study raise an important question: Are hormonal differences between male and female underlying sex-specific differences in $K_V2.1$ and $Ca_V1.2$ organization and function? Although our data do not provide an answer to this question, an intriguing approach would be to investigate the impact of ovariectomy and castration in mice and the role of $Ca_V1.2$ and $K_V2.1$ channel clustering and gating in arterial smooth muscle function. These experiments would help to understand whether estrogen or testosterone are responsible for the sex-specific variations in $Ca_V1.2$ and $K_V2.1$ channels reported here. For instance, would ovariectomy result in more "male-like" myocytes with smaller $K_V2.1$ and $Ca_V1.2$ clusters or the opposite for castrated males?

The work contained here hints at several broad implications in vascular physiology. From a general point of view, our data suggest that modulation of the phosphorylation state of $K_V2.1$ channels influence their capacity to form macro-clusters. Increased phosphorylation of $K_V2.1$ channels at serine 590 can lead to $K_V2.1$ macro-clustering promoting the formation of larger $Ca_V1.2$ channel clusters. This, in turn, increases the probability of cooperative gating of these channels and hence increases $Ca^{2+}$ influx, $[Ca^{2+}]_i$, and myogenic tone altering arterial diameter and arterial blood flow. In principle, this could happen in male and female myocytes and represents a novel mode of regulation of $Ca^{2+}$ influx in these cells.

A recent paper by Hernandez-Hernandez et al.[31] describes a computational model of mesenteric smooth muscle which incorporated electrophysiology and $Ca^{2+}$ signaling data to study sex-specific differences in $Ca_V1.2$ and $K_V2.1$ channel function. This study yielded results that suggests female myocytes may be more sensitive to $Ca^{2+}$ channel blockers compared to male. Interestingly, several clinical studies[52–54] have observed increased responses to $Ca^{2+}$ channel blockers in hypertensive women compared to men. For example, Kloner et al.[52], demonstrated a significantly larger proportion of women (91.4%) reached the goal diastolic blood pressure response compared to men (83%) when subjected to dihydropyridine-type channel blockers like amlodipine or nifedipine. Notably, following adjustments for potential confounding variables such as baseline blood pressure, age, weight, and dosage per kilogram, the differences remained significant. A more recent study by van Luik et al.[54], corroborated these findings, showing that treatment with calcium channel blockers led to a substantial reduction in systolic blood pressure, diastolic blood pressure, and mean arterial pressure in both sexes, with a notably more significant decrease observed in females compared to males. While many factors could be at play, including whether our mouse data fully translates to humans, our data, when considered alongside modeling and clinical research, prompt an interesting hypothesis. This data suggests the hypothesis that higher $K_V2.1$ macro-clustering, and hence $Ca_V1.2$ channel clustering and $Ca^{2+}$ influx, in arterial myocytes could be responsible for the increased responsiveness to these drugs in women than in men. However, future studies are warranted.

To conclude, we propose a model by which $K_V2.1$ serves a structural role in promoting $Ca_V1.2$ channel clustering and activity in a sex-dependent manner. Of note, $K_V2.1_{S590A}$ mutation reduced $Ca_V1.2$ clustering and function in female myocytes but had no effect on male myocytes. $K_V2.1$ clustering is not necessary for $K_V2.1$ channel function however, $K_V2.1$ macro-clusters alter $Ca_V1.2$ channel organization. Together, our data suggest that the interactions between $K_V2.1$ and $Ca_V1.2$ are crucial for sex-based differences in arterial smooth muscle physiology.

## Methods

### Generation of the CRISPR/Cas9-edited $K_V2.1_{S590A}$ (KCNB1 S590A) knock-in mouse.
The KCNB1 S590A mutation changes a AGC codon to GCC in Exon 2, thus converting a serine to an

alanine (S590A) in the $K_V2.1$ polypeptide. The knock-in mouse was generated in collaboration with the UC Davis Mouse Biology Program by using CRISPR/Cas mediated homology directed repair. KCNB1 S590A mice were generated by introducing a mixture of gRNA (15 ng/L), single-stranded oligodeoxynucleotide (ssODN) repair template and Cas9 protein (30 ng/µL) by pronuclear microinjection into C57BL/6 J (RRID: IMSR_JAX000664) mouse zygotes. Twenty zygotes were injected and implanted into the oviducts of one surrogate dam. A total of 6 pups were born, and genomic DNA was extracted from tail biopsies followed by PCR amplification using a specific primer set to identify a single male founder (F0). DNA-Seq analysis was used to confirm the mouse genotype. The correctly integrated single mutant F0 male mouse was further backcrossed with WT C57BL/6 J female mice to produce offspring (F1) followed by intercrossing for two additional generations to obtain KCNB1 S590A heterozygotes which were used for breeding. Heterozygous and homozygous mutants were identified by a PCR genotyping protocol.

**Animals.** All experiments were conducted in accordance with the University of California Institutional Animal Care and Use Committee guidelines. Animals were housed under standard light-dark cycles and allowed to feed and drink ad libitum. Mice were euthanized with a single, lethal dose of sodium pentobarbital (250 mg/kg) intraperitoneally.

**Arterial myocyte isolation.** Third- and fourth-order mesenteric arteries were carefully cleaned of surrounding adipose and connective tissue, dissected out, and placed in ice-cold dissecting solution containing (in mM) 5 KCl, 140 NaCl, 2 $MgCl_2$, 10 glucose, and 10 HEPES adjusted to pH 7.4 with NaOH. Arteries were first placed in dissecting solution supplemented with 1.23 mg/ml papain (Worthington Biochemical, LS003119) and 1 mg/ml DTT at 37 °C for 14 min. This was immediately followed by a five-min incubation in dissecting solution supplemented with 1.6 mg/ml collagenase H, 0.5 mg/ml elastase (Worthington Biochemical, LS002292), and 1 mg/ml trypsin inhibitor from *Glycine max* at 37 °C. Arteries were rinsed three times with dissection solution and single cells obtained by gentle trituration with a wide-bore glass pipette. Myocytes were maintained at 4 °C until used.

**HEK293T cell culture and transfection.** HEK293T (AATC, CRL-3216) cells were cultured in Dulbecco's Modified Eagle Medium (Gibco, 11955) supplemented with 10% fetal bovine serum (Gibco, 26140) and 1% penicillin/streptomycin (Gibco, 15140122) and maintained at 37 °C in a humidified 5% $CO_2$ atmosphere. Cells were transiently transfected using JetPEI (Polyplus Transfection, 101000053) according to manufacturer's protocol and passaged 24 h later onto 25 mm square #1.5 coverslips or 18 mm square collagen coated #1.5 coverslips (Neuvitro Corporation, GG-18-15-Collagen) for GSD experiments. Plasmids encoding DsRed-$K_V2.1_{WT}$, DsRed-$K_V2.1_{S586A}$, DsRed-$K_V2.1_{P404W}$, and DsRed-$K_V2.1_{P404W, S586A}$ were used for this study[17,21]. mScarlet-tagged versions of these plasmids were generated by GenScript, replacing the sequence encoding DsRed with sequence encoding mScarlet[55]. For the bimolecular fluorescence experiments, cells were transfected with the pore-forming subunit of the rabbit $Ca_V1.2$ (α1c, kindly provided by Dr. Diane Lipscombe; Brown University, Providence, RI) with the carboxy tail fused to either the N-fragment (VN) or the C-fragment (VC) of the Venus protein (Addgene, Cambridge, MA, 27097, 22011), auxiliary subunits $Ca_V\alpha_2\delta$, $Ca_V\beta_3$ (kindly provided by Dr. Diane Lipscombe, Brown University, Providence, RI) and either DsRed-$K_V2.1_{P404W}$ or DsRed-$K_V2.1_{P404W, S586A}$-DsRed. HEK293T cells were transfected with $Ca_V1.2$-VN, $Ca_V1.2$-VC, $Ca_V\alpha_2\delta$, $Ca_V\beta_3$ and DsRed-$K_V2.1$-dsRed in a 1.0:1.0:1.0:1.5:0.4 ratio.

**Live cell confocal imaging.** HEK293T cells transfected with 200 ng of mScarlet-$K_V2.1_{WT}$ or mScarlet-$K_V2.1_{S586A}$ and seeded onto 25-mm square 1.5 coverslips approximately 16 hours before experiments. Imaging was performed in Tyrode III solution consisting of (in mM) 140 NaCl, 5.4 KCl, 1 $MgCl_2$, 1.8 $CaCl_2$, 5 HEPES, and 5.5 glucose, pH 7.4 with NaOH. Cells were imaged with an Olympus Fluoview 3000 confocal laser-scanning microscope equipped with an Olympus Plan-Apochromat 60x oil immersion lens (NA = 1.40).

Stacks of images were analyzed using Imaris 10 (Andor, Belfast). Briefly, $K_V2.1$-associated mScarlet signal was mapped to x/y/z centroid co-ordinates in each image stack using the Spots tool. Spots were assigned to all signal surpassing a fixed signal threshold and restricted to puncta greater than 100 nm (x/y) and 150 nm diameter (z), such that any bright signal with a volume greater than two voxels was identified as a $K_V2.1$ cluster. 'Region Growing' was utilized with a fixed manual threshold to apply variable sizing to $K_V2.1$ Spots, in line with the volume and brightness of mScarlet puncta. Finally, the Cell segmentation function was used to estimate cell boundaries based on low-intensity mScarlet signal and obtain an approximate cell volume.

**$K_V2.1$ immunofluorescence immunocytochemistry.** Immunofluorescence labeling was performed on freshly dissociated arterial myocytes. Cells were left to adhere for one hour at room temperature prior to fixation, fixed with 4% formaldehyde (Electron Microscopy Sciences, 50980487) diluted in phosphate-buffered saline (PBS) (Fisher Scientific, Hampton, NH) for 15 min at room temperature, washed, and incubated with 50 mM glycine (BioRad, 1610718) for 10 min to reduce aldehydes. The surface membrane was stained with wheat germ agglutinin (WGA) Alexa Fluor 488 (1 µM, ThermoFisher, W11261) for 10 minutes at room temperature followed by washing. Cells were then incubated in blocking buffer made of 3% w/v bovine serum albumin and 0.25% Triton X-100 in PBS, followed by incubation with mouse anti-Kv2.1 (mAb K89/34; RRID: AB_2877280; NeuroMab, Davis, CA, 1:200) diluted in blocking buffer for one hour at room temperature or overnight at 4 °C. Myocytes were washed, incubated at room temperature for one hour with Alexa Fluor 647-conjugated donkey anti-mouse IgG diluted in blocking buffer (2 µg/ml, Molecular Probes, A31571) followed by washes in PBS. For experiments investigating $K_V2.1$ phosphorylation state, double labeling was performed with the mouse anti-$K_V2.1$ pS590 phosphospecific mAb L100/1[30] together with rabbit anti-$K_V2.1$ (KC[23]; Trimmer laboratory, RRID:AB_2315767; 1:100). Myocytes were washed, incubated at room temperature for one hour with Alexa Fluor 568-conjugated goat anti-mouse IgG (2 µg/ml, Molecular Probes, A11004) and Alexa Fluor 647-conjugated donkey anti-rabbit IgG (2 µg/ml, Molecular Probes, A31571) diluted in blocking buffer followed by washes in PBS. All washes were performed with PBS three times for 10 minutes. Coverslips were mounted onto microscope slides in Vectashield mounting medium (Vector Labs) and sealed with clear nail polish. Images were collected on a Dragonfly 200 spinning disk confocal (Andor), coupled to a DMi* Leica microscope (Leica, Wetzlar, Germany) equipped with a 60x oil immersion objective (NA = 1.40) and acquired using an Andor iXon EMCCD camera. Images were collected via Fusion software, in optical planes with a z-axis of 0.13 µm/step.

Image stacks were segmented and analyzed in Imaris 10. WGA-488 signal was background-subtracted and a fixed threshold applied to consistently map the plasma membrane, using the

Surfaces tool. Alexa Fluor-647 signal (denoting $K_V2.1$ puncta) was assessed using the Spots tool, as described above. Spots marking $K_V2.1$ clusters were categorized into internal and plasma membrane-restricted components, with the latter utilized for analysis.

**Super-resolution microscopy**. HEK293T cells transfected with 200 ng mScarlet-Kv2.1$_{WT}$ or mScarlet-Kv2.1$_{S586A}$-mScarlet and arterial myocytes were plated onto collagen coated glass coverslips (Neuvitro Corporation, GG-18-1.5-Collagen) followed by fixation with 3% formaldehyde and 0.1% glutaraldehyde diluted in PBS for 15 min at room temperature. After washing with PBS, cells were incubated with 50 mM glycine for 10 min to quench aldehydes. Cells were washed and incubated for one hour at room temperature with a blocking buffer made with 3% w/v BSA and 0.25% Triton X-100 in PBS. Cells were then incubated with either mouse anti-Kv2.1 (HEK293T experiments, mAb K89/34; RRID: AB_2877280; UC Davis/NIH Neuromab Facility, Davis, CA; 1:20) or mouse anti-Ca$_V$1.2 (arterial myocytes experiments, mAb L57/23; RRID: AB_2802123; 1:5). After extensive washings with PBS (three quick washes followed by three 30-min washes), cells were incubated at room temperature for one hour with Alexa Fluor 647-conjugated goat anti-mouse diluted in blocking buffer to a concentration of 2 μg/ml and afterwards extensively washed with PBS.

The imaging buffer contained 10 mM MEA, 0.56 mg/ml glucose oxidase, 34 μg/ml catalase, and 10% w/v glucose in TN buffer (200 mM Tris-HCl pH 8, 10 mM NaCl). A super resolution ground state deletion system (SR-GSD, Leica, Wetzlar, Germany) based on stochastic single-molecule localization was used to generate super-resolution images of Ca$_V$1.2 and $K_V2.1$ labeling. The Leica SR-GSD is a Leica DMI6000B TIRF microscope system equipped with a 160x HCX Plan-Apochromat (NA 1.43) oil-immersion lens and an EMCCD camera (iXon3 897, Andor Technology, Belfast, United Kingdom). Fluorophores were excited with a 642 nm laser (used for both pumping to the dark state and image acquisition). For all experiments, the camera was running in frame-transfer mode at a frame rate of 100 Hz (11 ms exposure time). Fluorescence was detected through Leica high power TIRF filter cubes (488 HP-T, 532 HP-T, 642 HP-T) with emission band-pass filters of 505–605 nm, 550–650 nm, and 660–760 nm. A total of 35,000 images were collected per cell and used to construct the super resolution localization images. Fluorescence signals in each image were fit with a 2D Gaussian function which localized the coordinates of centroids of single molecule fluorescence within the LASAF software (Leica). Images were rendered at 20 nm/pixel (normalized Gaussian mode), threshold (# photons/event) using the GSD software and exported as binary TIF images. Particle analyses were determined in ImageJ. Representative images were rendered down to 2 nm for visualization purposes.

To accomplish the Gaussian blur, the GSD generated pixel in an image was replaced by a weighted average of 200 nm of its neighboring pixels. The amount of blur applied to the image was controlled by the size of the kernel, which determines the radius of the neighboring pixels used in the calculation, such that the larger the kernel, the more pixels are included in the calculation, and the stronger the blur effect.

**Quantitative PCR**. Total RNA was isolated using the RNeasy Mini Kit (Qiagen) as per manufacturer's instructions. Isolated mRNA was then reverse transcribed using the AffinityScript qPCR cDNA Synthesis Kit (Agilent 600559) following manufacturer's protocol. Quantitative PCR (qPCR) analysis was performed using a QuantStudio 7 Pro Real-time PCR System (Applied Biosystems), using PowerUP SYBR Green Master Mix (Thermo Fisher Scientific, A25742) as the fluorescence probe. The cycling conditions were 50 °C for 2 minutes and 95 °C for 10 min, followed by 40 cycles of 95 °C for 15 s and 56 °C for 1 min. A dissociation curve protocol (ramping temperatures between 60 °C and 95 °C) was added at the end to verify amplification specificity of each qPCR reaction.

Specific primers were designed in this experiment, including β-actin (NM_007393.5): sense nt (895-914): CCAGCCTT CCTTCTTGGGTA, antisense nt (989-967): AGAGGTCTTT ACGGATGTCAACG; and Ca$_V$1.2 (NM_009781.4): sense nt (5-23): CTGAAAGCAGAAGCTCGGA, antisense nt (181-163): CATTGTGGCTTCCAGTTGG. Primer efficiencies were tested to be in between 90% and 110%. The relative abundance of Ca$_V$1.2 transcript was normalized to β-actin transcript expression.

**Proximity ligation assay**. A Duolink In Situ PLA kit (Sigma DUO92007) was used to detected $K_V2.1$-$K_V2.1$ and $K_V2.1$-Ca$_V$1.2 complexes in freshly isolated mesenteric arterial myocytes. All protocols post incubation of primary antibodies were followed in accordance with the manufacturer's instructions. Briefly, cells were plated on glass coverslips and allowed to adhere for 1 h at room temperature. Cells were fixed with 4% paraformaldehyde for 20 min, quenched in 10 mM glycine for 15 min, washed in PBS two times for three minutes, and permeabilized 20 min in 0.1% Triton X-100. After blocking for 1 h at 37 °C in Duolink Blocking Solution, cells were incubated overnight at 4 °C using the following primary antibodies: mouse anti-$K_V2.1$ (mAb K89/34; RRID: AB_2877280; UC Davis/NIH Neuromab Facility, Davis, CA; 1:200), rabbit anti-$K_V2.1$ (KC[23]; RRID:AB_2315767; 1:100) and rabbit anti-Ca$_V$1.2. The anti-Ca$_V$1.2 rabbit polyclonal antibody "Ca$_V$1.2 II-III" was generated by immunizing two New Zealand white rabbits with a His-tagged recombinant protein fragment corresponding to a.a. 785-900 of mouse Ca$_V$1.2 (accession number Q01815). Antibodies were affinity purified from serum on nitrocellulose strips containing the Ca$_V$1.2 II-III His-tagged recombinant protein fragment following the method of Olmsted[56]. Cells incubated with only one primary antibody served as negative controls. $K_V2.1^{-/-}$ null mice also served as negative biological controls as previously reported[15]. Secondary oligonucleotide-conjugated antibodies (PLA probes: anti-mouse MINUS and anti-rabbit PLUS) were used to detect $K_V2.1$ and Ca$_V$1.2 interactions. Fluorescent signal was detected using an Olympus FV3000 confocal microscope equipped with a 60x oil immersion lens (NA = 1.40). Images were collected with a z-axis of 0.5 μm/step optical planes. Stacks of images were combined in ImageJ and used for analysis of puncta/μm$^2$ per cell restricting the PLA signal to the edge of the cell.

**Patch-clamp electrophysiology**. All electrophysiological recordings were acquired at room temperature using an Axopatch 200B amplifier and Digidata 1440 digitizer (Molecular Devices, Sunnyvale, CA). Borosilicate patch pipettes were pulled and polished to resistances of 3–6 MΩ using a micropipette puller (model P-97, Sutter Instruments, Novato, CA).

$I_{Kv2.1}$ was measured in arterial myocytes using conventional whole-cell voltage-clamp electrophysiology at a frequency of 50 kHz and low-pass filter of 2 kHz. Cells were continuously perfused with an external solution containing (in mM) 130 NaCl, 5 KCl, 3 MgCl$_2$, 10 glucose, and 10 HEPES adjusted to pH 7.4 with NaOH. Micropipettes were filled with an internal solution containing (in mM) 87 K-aspartate, 20 KCl, 1 CaCl$_2$, 1 MgCl$_2$, 5 MgATP, 10 EGTA, and 10 HEPES pH 7.2 with KOH. A liquid junction potential of 13 mV was corrected for offline. To measure current-voltage relationships, cells were subjected to a series of

500 ms test pulses increasing from $-70$ mV to $+70$ mV. In order to isolate the RY785-sensitive Kv2 current, cells were first bathed and recorded in external solution. Cells were then exposed to 1 μM RY785 (MedChemExpress, HY-114608) to inhibit Kv2.1 activity. RY785-sensitive currents were calculated by subtracted the RY785 exposed traces from the composite $I_K$ traces.

$I_{Ca}$ was measured in isolated arterial myocytes using conventional whole-cell electrophysiology. Currents were measured at a frequency of 50 kHz and low-pass filtered at 2 kHz. Myocytes were continuously bathed in an external solution with (in mM) 115 NaCl, 10 TEA-Cl, 0.5 MgCl$_2$, 5.5 glucose, 5 CsCl, 20 BaCl$_2$, and 10 HEPES adjusted to a pH of 7.4 using CsOH. Micropipettes were filled with (in mM) 20 CsCl, 87 aspartic acid, 1 MgCl$_2$, 10 HEPES, 5 MgATP, and 10 EGTA adjusted to pH 7.2 via CsOH. A voltage error of 9.4 attributed to the liquid junction potential of the recording solutions was corrected for offline. Cells were exposed to a series of 300 ms depolarizing pulses from a holding potential of $-70$ mV to test potentials ranging from $-70$ mV to $+60$ mV to attain current-voltage relationships.

**Bimolecular fluorescence complementation**. Spontaneous interactions of Ca$_V$1.2 channels were assayed using biomolecular fluorescence complementation. HEK293T cells were transfected with Ca$_V$1.2 channels tagged at their C-terminus to either a non-fluorescent N-(VN(1-154, I152L)) or C-terminal (VC(155-238, A206K)) halves of a 'split' Venus fluorescent protein. When Ca$_V$1.2-VN and Ca$_V$1.2-VC are brought close enough together to interact, the full Venus protein can fold into its functional, fluorescent conformation. The scale of Venus fluorescence emission can therefore be an indicator of Ca$_V$1.2-Ca$_V$1.2 interactions. Venus fluorescence was monitored using TIRF microscopy.

For whole-cell current recordings from HEK293T cells, pipettes were filled with a solution containing (mM) 84 Cs-aspartate, 20 CsCl, 1 MgCl$_2$, 10 HEPES, 1 EGTA, and 5 MgATP adjusted to pH 7.2 using CsOH. HEK293T cells were continuously perfused with an external solution comprising of (in mM) 5 CsCl, 10 HEPES, 10 glucose, 140 NMDG, 1 MgCl$_2$, and 20 CaCl$_2$ with a pH of 7.4 (HCl).

**Ca$_V$1.2 sparklets**. Ca$^{2+}$ sparklets were recorded using a through-the-lens TIRF microscope built around an inverted microscope (IX-70; Olympus) equipped with a Plan-Apochromat (60x; NA 1.49) objective (Olympus) and an electron-multiplying charge-coupled device camera (iXON; Andor Technology, UK). Myocytes were loaded via the patch pipette with a solution containing (in mM) 0.2 Fluo-5F (Invitrogen, F14221), 87 Cs-aspartate, 20 CsCl, 1 MgCl$_2$, 5 MgATP, 10 HEPES, 10 EGTA, pH 7.2 with CsOH. Cells were perfused in an external solution containing 140 NMDG, 5 CsCl, 51 MgCl$_2$, 10 glucose, 10 HEPES, 2 CaCl$_2$, pH 7.4 with HCl. After obtaining a GΩ seal in 2 mM Ca$^{2+}$ external solution, the cell was broken into and allowed to dialyze for 3 min. The external solution was exchanged with a solution containing (in mM) 120 NMDG, 5 CsCl, 1 MgCl$_2$, 10 glucose, 10 HEPES, 20 CaCl$_2$, pH 7.4 with HCl. Images for the detection of sparklets were recorded at a frequency of 100 Hz using TILL Image software. Cells were held to a membrane potential of $-70$ mV using the whole-cell configuration of the patch-clamp technique. Sparklets were automatically detected and later analyzed using custom software (Source code 2) written in MATLAB (RRID:SCR_001622)[10].

**In silico modeling**. Simulations were performed using the Sato et al.[9] and Hernandez-Hernandez et al.[31] mathematical models

for cluster formation and smooth muscle electrophysiology, respectively.

**Statistics and reproducibility**. Statistical testing was performed using GraphPad Prism (GraphPad Software, Inc.). All data were tested for normality using Shapiro-Wilk test. Data are expressed as mean ± standard error of the mean (SEM). Data were compared using one tailed unpaired $t$-tests when comparing two groups or one-way analysis of variance (ANOVA) with a Tukey post-hoc test for more than two groups. $P < 0.05$, denoted by * in the figures, was considered statistically significant and further classified as $P < 0.01$ (**), $P < 0.001$ (***), $P < 0.0001$ (****).

**Chemicals**. All chemical reagents were acquired from Sigma-Aldrich (St. Louis, MO) unless otherwise stated.

**Reporting summary**. Further information on research design is available in the Nature Portfolio Reporting Summary linked to this article.

## Data availability
All source data are available in the Supplementary Data file.

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

## Acknowledgements

We thank Dellaney Rudolph-Gandy and Ernesto Javier Rivera for technical assistance. We thank Dr. Joshua Tulman for assistance with the diagram. This work was supported by grants from US National Institutes of Health HL085686 (L.F.S.), US National Institutes of Health HL128537 (L.F.S., C.E.C.), US National Institutes of Health HL144071 (L.F.S., J.S.T.), US National Institutes of Health NS114210 (L.F.S., J.S.T.), US National Institutes of Health 1OT2OD026580 (L.F.S., C.E.C.), Amazon AWS Cloud Credits for Research (G.H.H.).

## Author contributions

Conceptualization: C.M., L.F.S., J.S.T. Methodology: C.M., L.F.S., N.C.V., J.S.T. Investigation: C.M., S.C.O., D.M., G.H.H., P.R., Z.F., D.S. Formal Analysis: C.M., S.C.O., D.M., P.R., Z.F., L.F.S. Visualization: C.M. Data Curation: G.H.H., D.S. Software: G.H.H., D.S. Supervision: L.F.S., C.E.C., J.S.T. Writing—original draft: C.M., L.F.S. Writing—review & editing: C.M., S.C.O., D.M., G.H.H., P.R., Z.F., N.C.V., J.S.T., L.F.S.

## Competing interests

The authors declare no competing interests.
