## [Peer Review File · Communications Biology]

Reviewers' comments:

Reviewer #1 (Remarks to the Author):

In this manuscript, Matsumoto and colleagues investigate the role of Kv2.1 channels in controlling Cav1.2 clustering and therefore activity. Overall, the draft is outstanding. The authors claim that the degree of clustering of Kv2.1 dictates Cav1.2 activity in a sex-dependent manner. I believe that the paper will influence the field. The authors intriguingly show that the expression of Kv2.1 alone is insufficient to affect Cav1.2 activity. They also provide compelling evidence using native smooth muscle cells showing a structural role for Kv2.1 that ultimately controls Cav1.2 activity. I only have a few comments and suggestions for the authors to consider.

- Sex differences: The authors provide a convincing mechanism underlying the differences between sexes. I am curious to know the authors' thoughts on the origin of these differences. Are these hormonal influences?

- Figure 5: Panels A and B show Cav1.2 currents evoked by a voltage step to 0 mV (?). The figure included a step protocol to +50 mV, even though the traces are at 0 mV (I think). Please, check.

- A schematic diagram at the end would be very useful.

Reviewer #2 (Remarks to the Author):

The formation of KV2.1 macro-clusters is required for sex-specific differences in L-type CaV1.2 clustering and function in arterial myocytes

Matsumoto et al. (Santana)

In what amounts to part 2 of this group's previous paper (O'Dwyer et al. 2020), Matsumoto et al. drill down into the previously introduced idea that KV2.1 protein serves a sex-specific structural role in enhancing CaV1.2 clustering in addition to its conducting function. Using a battery of experimental approaches, including exogenous expression systems, superresolution (TIRF) microscopy, electrophysiological techniques and a novel gene-edited knock-in mouse, the authors demonstrate that KV2.1 forms micro-clusters that are transformed into macro-clusters by phosphorylation of a key serine residue (S586 and S590 in mouse and rat KV2.1, respectively). They further show that KV2.1 phosphorylation at S590 is enhanced in female myocytes/constitutively lower in male myocytes; as a result, clustering activity (volume, density, membrane occupancy) is higher in female than male myocytes. S590A mutation (KV2.1S590A-knock-in mice) de-clusters KV2.1 channels selectively in female myocytes, but notably does not affect KV2.1 conducting activity. In addition, they demonstrate that KV2.1 de-clustering results in sex-specific disruption of CaV1.2 channel regulation, reducing KV2.1-CaV1.2 clustering, CaV1.2 cluster size, and macroscopic CaV1.2 currents in female, but not male, myocytes (KV2.1S590A-knock-in mice), while also decreasing CaV1.2-CaV1.2 interactions and CaV1.2 sparklets (exogenous expression of KV2.1S586A). Essentially, KV2.1 channel de-clustering converts female myocytes to male myocytes. The methods deployed are appropriate, the data are of high quality, and the conclusions follow logically from the results. Unanswered questions – Why is KV2.1S590A phosphorylation constitutively lower in males? What physiological ends do the observed sex-specific differences serve? – suggest additional chapters to this story.

Comments/questions

The authors propose (abstract) "that the degree of KV2.1 clustering tunes CaV1.2 channel function in a sex-specific manner in arterial myocytes." To what end? What distinguishable differences between male and female physiology arise from the observed sex-specific difference in expression and CaV1.2 cluster formation-induced behavior of KV2.1 in vascular myocytes? In other words, what are the "sex-based differences in arterial smooth muscle physiology" that these functional differences account for? The authors are encouraged to speculate about how their findings fit into the larger context of vascular physiology.

How is it that mean KV2.1 cluster area is significantly different between HEK293T cells exogenously transfected with KV2.1 versus KV2.1S586A (Fig. 1H) but mean cluster volume is not (Fig. 1E)? This seems mathematically impossible.

Figure S3C shows a substantial decrease in KV2.1–KV2.1 PLA puncta density, but the corresponding panel depicting CaV1.2-KV2.1 PLA puncta data (Figure S3F) shows a rather modest decrease in the number of puncta per unit area and considerable variability. The represented individual cells from female KV2.1WT and KV2.1S590A myocytes depicted in S3D (bottom) leave the impression of fewer, but larger, CaV1.3–KV2.1 puncta. Is this visual impression typical for other representative examples? What would a quantitative analysis like that presented in Figure 1 for KV2.1–KV2.1 clusters show for CaV1.3-KV2.1 puncta size in KV2.1WT and KV2.1S590A myocytes?

Are all KV2.1 clusters that are included in analyses localized to the sarcolemma, or are these a mix of plasma membrane and intracellular clusters? The statement that “confocal images of [WT and mutant] myocytes subjected to PLA show that puncta of KV2.1-KV2.1 PLA signals were randomly distributed throughout the cell” (Suppl. Fig. 3) would suggest a mixture. (However, the phrase “throughout these cells” was also used to describe the distribution of clusters based on TIRF microscopy, which focuses – literally – on plasmalemmal clusters.) If the clusters that are analyzed are those in the plasmalemma and not a mix of plasma membrane and intracellular clusters, please clarify this in the manuscript. If they do include both intracellular and plasma membrane-localized clusters, where are these intracellular clusters localized, and what is their relevance to the question at hand? How would excluding them from the analysis affect the conclusions reached?

Data presented in Figure 8 depicting CaV2.1 sparklet activity imply a relative dearth of high-affinity sites in males ($n = 1$ event in WT males and apparently zero such events in males during the recording window in KV2.1S590A myocytes). Since these high-affinity sites are those that are meaningful in terms of increasing intracellular Ca²⁺, what does this say about the basic functionality of male and female myocytes? From a naïve perspective, it seems like male myocytes are perched perilously close to a largely non-functional state.

Minor points

The header for Suppl. Fig. 3 implies a decrease in KV2.1 and CaV2.1 channels in female KV2.1S590A myocytes, rather than a decrease in their interaction, and should be rephrased.

Reviewer #3 (Remarks to the Author):

The authors investigated Kv2.1 channel clusters forming roles in Cav1.2 coupling and propose a sex-specific functional dependency role for these clusters in arterial myocytes. They suggest that phosphorylation of S590 is required for altering the micro-clusters to form macro-clusters and the degree of this phosphorylation was found to be more enhanced in female arterial myocytes. Disrupting this site by mutagenesis averted such transformation and resulted in changes in Cav1.2 cluster size and activity when compared to male arterial myocytes. The authors used a multi-disciplinary approach; molecular biology, in vitro calcium imaging, immunofluorescence staining, super-resolution microscopy, in situ detection of protein interactions, and patch clamp electrophysiology. In silico modeling simulations were utilized to confirm their finding. Although Kv2.1 channel activity seems unchanged, distinct, and independent from these large clusters, the authors suggest they are dynamically associated with Cav1.2, thus regulating its function explicitly in female arterial myocytes.

The authors provide p values and a general description of statistical tests used is described in methods, however, they do not specify which statistical test/tests were used to generate the significant levels obtained in the figures.

Except for 'n' indicating the number of cells used for Figure 1 (line 14) I could not find the 'n' values for the other experiments/ replica shown in the presented figures.

Please can the authors clarify the following queries:

1. What is the ratio of S590 phosphorylation to the unphosphorylated state?
2. Please provide RR Identifier number for C57BL/6J strain (line 71).

3. Would it be possible to use co-immunoprecipitation to identify the physiologically relevant Kv1.2–Cav1.2 interactions? Rather than identifying the distribution patterns described in Supplemental Figure 3. For example, Supplemental Figure 3F shows that the number of points on the bar chart (Puncta/ μm^2) for WT is more than double (40 points) for that of S590A (18 points). Are these shown points based on 1 cell or do they each present different cells? If so why is the WT count double? It's also unclear why the authors did not use KV2.1-/- null animals for these experiments. As it's shown from Supplemental Figure 2 that these animals are available for this study.

COMMSBIO-23-2290

We thank the Editor and Reviewers for their thoughtful comments on our work. In the revised version of the manuscript, we included new data, analyses, and text suggested by the Reviewers. The inclusion of the new material strengthened the conclusions of the study and enhanced its potential impact in the field. We hope the paper is now acceptable for publication in *Communications Biology*.

Response to comments by Reviewer #1:

- *"In this manuscript, Matsumoto ad colleagues investigate the role of Kv2.1 channels in controlling Cav1.2 clustering and therefore activity. Overall, the draft is outstanding."*

We thank the reviewer for their generous comment.

- *"Sex differences: The authors provide a convincing mechanism underlying the differences between sexes. I am curious to know the authors' thoughts on the origin of these differences. Are these hormonal influences?"*

We thank you for raising this important question. We too acknowledge the importance of sex differences and hormonal influences in the context of vascular physiology. Currently, we have an ongoing project where we are comparing myogenic tone between castrated, ovariectomized, and wild type mice. Our preliminary data suggest that it is testosterone, not estrogen, responsible for the sex-differences in Kv2.1 and Cav1.2 channel organization and function as well as myogenic tone in mesenteric arterial myocytes. Given the complexity and number of experiments to unveil the mechanistic details by which testosterone causes these differences, we believe a separate follow up paper should include the new data.

That said, we added a section in our discussion that recognizes this gap in knowledge and suggest potential mechanisms responsible for sex-specific differences in Kv2.1 and Cav1.2 clustering and activity (Discussion section, page 17, lines 802-808)

- *"Figure 5: Panels A and B show Cav1.2 currents evoked by a voltage step to 0 mV (?). The figure included a step protocol to +50 mV, even though the traces are at 0 mV (I think). Please, check."*

Thank you for bringing this mistake to our attention. The figure has been corrected to show that Cav1.2 currents were evoked by a voltage step to 0 mV.

- *"A schematic diagram at the end would be very useful."*

We thank you and agree a schematic diagram would be helpful. We have included the figure below as **Figure 9** (page 15)

Response to comments to Reviewer #2:

- *"The authors propose (abstract) 'that the degree of KV2.1 clustering tunes CaV1.2 channel function in a sex-specific manner in arterial myocytes.' To what end?"*

We thank you for this opportunity for clarification of this important point. To address this comment, we modified the abstract (page 1, lines 31-32) and discussion sections (page 15,

lines 687-703) of the manuscript to underscore the broader implications of our work on vascular function.

- “What distinguishable differences between male and female physiology arise from the observed sex-specific difference in expression and Ca_V1.2 cluster formation-induced behavior of KV2.1 in vascular myocytes?”

We added a paragraph in our Discussion section of the paper to address this specific comment (page 17, lines 790-800).

That said, for the benefit of the reviewers, we would like to mention that we are currently examining the role of sex hormones on wild type and K_V2.1_{S590A} smooth muscle for a follow up paper. Although experiments are still on going, preliminary results indicate that expression of K_V2.1_{S590A} decreases myogenic tone in female but not male arteries. These data are consistent with the data showing that expression of the clustering deficient K_V2.1_{S590A} channel decreases Ca_V1.2 clustering and function in female but not male myocytes included in the current manuscript.

- “The authors are encouraged to speculate about how their findings fit into the larger context of vascular physiology.”

Prompted by this comment, we added a paragraph in the Discussion section of the paper where we consider the larger implications for vascular physiology (page 17-18, lines 810-833).

- “How is it that mean KV2.1 cluster area is significantly different between HEK293T cells exogenously transfected with KV2.1 versus KV2.1S586A (Fig. 1H) but mean cluster volume is not (Fig. 1E)? This seems mathematically impossible.”

We agree that the mean cluster areas should be lower in K_V2.1_{S586A} HEK293T cells compared to K_V2.1_{WT} HEK293 cells. The reason for the apparent discrepancy is that our initial analysis erroneously rejected many of the small clusters which skewed our data. We apologize for this error.

Accordingly, we re-analyzed our images and found that K_V2.1_{S586A} transfected HEK293T cells expressed K_V2.1 clusters that were smaller compared to K_V2.1_{WT} transfected cells. Furthermore, we show that the total clusters per cell increased with S586A mutation yet the total volume per cell decreased. Effectively, the S586A mutation broke apart these large macro-clusters into smaller clusters. We updated the Result section to reflect this (updated Figure 1, page 4; updated text, page 3, lines 114-121).

“Figure S3C shows a substantial decrease in KV2.1–KV2.1 PLA puncta density, but the corresponding panel depicting Ca_V1.2-KV2.1 PLA puncta data (Figure S3F) shows a rather modest decrease in the number of puncta per unit area and considerable variability.”

Correct. The proximity ligation assay (PLA) reports, in the form of puncta, K_V2.1 channels that are within about 50 nm¹ from each other. Accordingly, K_V2.1-K_V2.1 PLA is a readout of intra-cluster channel-to-channel proximity, or an indication of K_V2.1 clustering itself. By contrast, Ca_V1.2-K_V2.1 PLA puncta indicate sites of inter-cluster proximity. This is now clarified in the manuscript (page 17, line 773-780).

- “The represented individual cells from female KV2.1WT and KV2.1S590A myocytes depicted in S3D (bottom) leave the impression of fewer, but larger, CaV1.3–KV2.1 puncta. Is this visual impression typical for other representative examples? What would a quantitative analysis like that presented in Figure 1 for KV2.1–KV2.1 clusters show for CaV1.3-KV2.1 puncta size in KV2.1WT and KV2.1S590A myocytes?”

We agree and apologize that our representative images can be misleading and bring the reader to this conclusion. We included new data and representative images that suggests that there are more Ca_v1.2-K_v2.1 puncta in K_v2.1_{WT} females compared to K_v2.1_{S590A} females. Additionally, we quantified the total Ca_v1.2-K_v2.1 puncta area per cell (i.e., sum of all puncta areas per cell) and found that these puncta occupied less area in K_v2.1_{S590A} females. This has been reflected in the supplemental results (Supplemental Figure 3).

- “Are all KV2.1 clusters that are included in analyses localized to the sarcolemma, or are these a mix of plasma membrane and intracellular clusters? The statement that “confocal images of [WT and mutant] myocytes subjected to PLA show that puncta of KV2.1-KV2.1 PLA signals were randomly distributed throughout the cell” (Suppl. Fig. 3) would suggest a mixture. (However, the phrase “throughout these cells” was also used to describe the distribution of clusters based on TIRF microscopy, which focuses – literally – on plasmalemmal clusters.) If the clusters that are analyzed are those in the plasmalemma and not a mix of plasma membrane and intracellular clusters, please clarify this in the manuscript. If they do include both intracellular and plasma membrane-localized clusters, where are these intracellular clusters localized, and what is their relevance to the question at hand? How would excluding them from the analysis affect the conclusions reached?”

We thank you for bringing this to our attention. For all confocal microscopy analyses in the main figure, we used membrane marker WGA488 to delineate and restrict the analysis to membrane associated K_v2.1 signal. While our PLA experiments were conducted without membrane staining, we restricted the PLA signal analyzed to the edge of our cells which based on arterial myocyte dimensions we are confident coincides with the sarcolemma of arterial myocytes. We have clarified this in the Methods section of the paper (page 20, lines 937-941 and pages 21-22, lines 1011-1013).

- “Data presented in Figure 8 depicting CaV2.1 sparklet activity imply a relative dearth of high-affinity sites in males (n = 1 event in WT males and apparently zero such events in males during the recording window in KV2.1S590A myocytes). Since these high-affinity sites are those that are meaningful in terms of increasing intracellular Ca²⁺, what does this say about the basic functionality of male and female myocytes? From a naïve perspective, it seems like male myocytes are perched perilously close to a largely non-functional state.”

We now include additional Ca_v1.2 sparklet data in this figure. It is important to note that we tried over 20 cells from multiple S590A knock in male and female mice, but despite our best efforts we only detected 1 high activity site in these cells. The nP_s of this sole high activity site was similar to that of wild type cells.

In the first version of the manuscript, we included references to the averages of the detectable high activity sites in the manuscript. We now incorporate in the analyses and figure data from cells where no sparklet activity could be detected and make note of this in the manuscript. Although this does not change the conclusions of the study, we more clearly note that the

overall sparklet activity within $K_{V2.1WT}$ males, $K_{V2.1S590A}$ males, and $K_{V2.1S590A}$ female myocytes are reduced compared to $K_{V2.1WT}$ females. (updated Figure 8, page 14; updated text pages 13-14, lines 612-661).

Minor points:

- “The header for Suppl. Fig. 3 implies a decrease in $KV2.1$ and $CaV2.1$ channels in female $KV2.1S590A$ myocytes, rather than a decrease in their interaction, and should be rephrased.”

Thank you for this comment. We have corrected the title to specifically reference the decrease in $K_{V2.1}$ - $Ca_{V1.2}$ proximity (Supplemental Figure 3).

Response to comments by Reviewer #3:

- “The authors provide p values and a general description of statistical tests used is described in methods, however, they do not specify which statistical test/tests were used to generate the significant levels obtained in the figures.”

We apologize for this omission. We have included this under our “Chemicals and statistics” header of our method section (page 23, lines 1076-1080).

- “Except for ‘ n ’ indicting the number of cells used for Figure 1 (line 14) I could not find the ‘ n ’ values for the other experiments/ replica shown in the presented figures.”

We now include “ n ” values for all experiments in the figure legends (All Main and Supplemental Figure legends).

- “What is the ratio of S590 phosphorylation to the unphosphorylated state?”

This is a very interesting question. This is a question we would love to have the answer to; however, because we rely on two different antibodies, simply dividing the amplitude of one signal to the other, is not likely to yield an accurate and hence meaningful number. We hope to address this important issue in follow up studies involving alternative approaches such as mass spectrometry.

- “Please provide RR Identifier number for C57BL/6J strain (line 71)”

Thank you for bringing this omission to our attention. We have included the RRID number for these mice IMSR_JAX:000664 to the materials and methods section (page 18, line 850).

- “Would it be possible to use co-immunoprecipitation to identify the physiologically relevant $Kv1.2$ – $Cav1.2$ interactions?”

Great suggestion.

First, we must first apologize for misrepresenting the results of our proximity ligation assay experiment as interactions. This characterization is incorrect as PLA is a readout of the proximity of two entities but not necessarily the direct physical interaction of the two. Hence, we changed all references to channel “interactions” to channel to channel “proximity” (Page 9-10, lines 392-450).

Second, we too agree that studying the potential physical interactions of $K_v2.1$ and $Ca_v1.2$ channels is of utmost importance. In the context of this paper, PLA is not the right tool to reveal that. To do so, we could include the use of co-immunoprecipitation experiments. One limitation is that co-IP would require $Ca_v1.2$ - $K_v2.1$ interactions to be stable, which we do not know. For co-IP to work, we need the interactions to be very stable, or could crosslink as a potential solution. We are interested in addressing this question but would like to do so in a follow up paper. In addition, we envision a combination of techniques to address this question that include but are not limited to our split-Venus bimolecular fluorescence system or FRET-based studies.

- “Rather than identifying the distribution patterns described in Supplemental Figure 3. For example, Supplemental Figure 3F shows that the number of points on the bar chart (Puncta/ μm^2) for WT is more than double (40 points) for that of S590A (18 points). Are these shown points based on 1 cell or do they each present different cells? If so, why is the WT count double?”

We are happy to clarify that each point in our bar charts correspond to one cell. We admit there was an omission of a data set in the $K_v2.1_{S590A}$ females which has now been included. There is a comparison of 40 cells in $K_v2.1_{WT}$ vs 30 in $K_v2.1_{S590A}$ group.

- “It’s also unclear why the authors did not use $KV2.1^{-/-}$ null animals for these experiments. As it’s shown from Supplemental Figure 2 that these animals are available for this study.”

Good suggestion. Our initial submission did not mention the use of $K_v2.1^{-/-}$ null mice for our negative control. However, we have previously published PLA data for $Ca_v1.2$ and $K_v2.1$ proximity in $K_v2.1^{-/-}$ null mice and found no signal². This experimental setup serves as an important negative control. Reference to this data is now included in the methods. (page 21, line 1007-1008)

References

- 1 Fredriksson, S. *et al.* Protein detection using proximity-dependent DNA ligation assays. *Nat Biotechnol* **20**, 473-477, doi:10.1038/nbt0502-473 (2002).
- 2 O'Dwyer, S. C. *et al.* Kv2.1 channels play opposing roles in regulating membrane potential, Ca(2+) channel function, and myogenic tone in arterial smooth muscle. *Proc Natl Acad Sci U S A* **117**, 3858-3866, doi:10.1073/pnas.1917879117 (2020).

REVIEWERS' COMMENTS:

Reviewer #1 (Remarks to the Author):

I'd like to thank the authors for addressing my comments. The manuscript has improved significantly after the authors addressed reviewers' comments. I have no additional comments.

Osama Harraz

Reviewer #2 (Remarks to the Author):

Great study and great job in the responses. Congratulations.

Reviewer #3 (Remarks to the Author):

The authors have adequately addressed my concerns and included required information in the new revised manuscript, I have no further comments.